# M-Sec promotes the production of infectious HIV-1 virus through the exocyst complex in macrophages

Reem M. Mahmoud[1,2], Masateru Hiyoshi[3], Randa A. Abdelnaser[1], Kazuaki Monde[4], Nami Monde[4], Takaaki Koma[5], Hidenobu Mizuno[6], Sara A. Habash[1,2], Naofumi Takahashi[1], Yosuke Maeda[4,7], Akira Ono[8], Shinya Suzu [1]*

1 Division of Infection & Hematopoiesis, Joint Research Center for Human Retrovirus Infection, Kumamoto University, Kumamoto, Japan, 2 Faculty of Medicine, Clinical Pathology, Suez Canal University, Ismailia, Egypt, 3 Center for Next-Generation Biologics Research, National Institute of Infectious Diseases, Japan Institute for Health Security, Tokyo, Japan, 4 Department of Microbiology, Faculty of Life Sciences, Kumamoto University, Kumamoto, Japan, 5 Department of Microbiology, Graduate School of Medicine, Tokushima University, Tokushima, Japan, 6 International Research Center for Medical Sciences, Kumamoto University, Kumamoto, Japan, 7 Department of Nursing, Kibi International University, Okayama, Japan, 8 Department of Microbiology and Immunology, University of Michigan Medical School, Ann Arbor, Michigan, United States of America

* ssuzu06@kumamoto-u.ac.jp

## Abstract

We have demonstrated that the cellular protein M-Sec promotes the transmission of human immunodeficiency virus type 1 (HIV-1) in macrophages. However, the underlying mechanism is not fully understood. Here, we report that M-Sec promotes the production of infectious HIV-1 virus. The major viral structural protein Gag distributed as many puncta in infected cells, which is one of the indicators of viral particle formation. The knockdown of M-Sec hindered the Gag puncta formation and co-localization of Gag with the viral envelope protein Env in cells, and reduced the amount of Env and infectivity of the produced virus. Consistent with these results, the overexpression of M-Sec induced the accumulation of Gag puncta, Gag/Env co-localization, and Env incorporation into virus and viral infectivity. M-Sec is known to bind phosphatidylinositol 4,5-bisphosphate (PIP2) and a small GTPase Ral, both of which were required for the M-Sec-mediated HIV-1 regulation. The exocyst complex, which is the downstream effector of Ral, was also required for the M-Sec-mediated HIV-1 regulation. Because PIP2, Ral and the exocyst complex are important for the M-Sec-mediated formation of the long plasma membrane protrusions, the present study suggests that M-Sec promotes HIV-1 transmission by acting on both cell structures and viral production through these overlapping components.

**Data availability statement:** All data generated and analysed in this study are included in the manuscript and supplemental figures. The minimal data set is available in a supporting file (S1 Raw Images).

**Funding:** This study was supported by a grant (KAKENHI) from the Japan Society for the Promotion of Science (JSPS) (23K27418 to SS), a grant from Takeda Science Foundation (to TK), and a grant from the Japan Agency for Medical Research and Development (AMED) Research Program on HIV/AIDS (JP24fk0410065h0001 to KM). AO is supported by an NIH grant R37 AI071727. The funders had no role in study design, data collection and analysis, decision to publish, or preparation of the manuscript.

**Competing interests:** The authors have declared that no competing interests exist.

## Author summary

Despite an effective anti-retroviral therapy, human immunodeficiency virus type 1 (HIV-1) persists in a fraction of infected cells, which is an obstacle to cure. HIV-1 exploits the cell-to-cell infection for its transmission, which is more efficient than infection by cell-free virus. Thus, it is important to fully understand the process of cell-to-cell infection towards the HIV-1 cure. We previously identified M-Sec as the cellular protein that potentiates the cell-to-cell infection of HIV-1. However, the underlying mechanism is not fully explained. In this study, we discovered that M-Sec promotes the production of infectious HIV-1 particles. Mechanistically, M-Sec affects the intracellular dynamics of the major viral structural protein Gag, which leads to an efficient incorporation of the viral envelope protein Env into viral particles. This activity of M-Sec depends on PIP2 (the phosphoinositide), Ral (the small GTPase), and the exocyst complex (the downstream effector of Ral), all of which are involved in vesicular trafficking. Thus, the present study identifies M-Sec and related molecular components as potential targets of anti-HIV-1 strategies.

## Introduction

M-Sec (a.k.a TNFAIP2), the TNF-α-inducible 74-kD cytoplasmic protein, has no catalytic activity but is known to regulate cell motility and structure. For example, M-Sec promotes the migration of breast cancer cells [1] and nasopharyngeal carcinoma cells [2]. Another well-known activity of M-Sec is the induction of tunneling nanotubes (TNTs) [3], the F-actin-containing long extensions or protrusions of the plasma membrane which often connect distant cells [4]. It is reasonable to speculate that these two activities of M-Sec are beneficial not only for invasion/metastasis of cancer cells but also for cell-to-cell infection of pathogens [5], because high motility and TNT formation of infected cells increase the likelihood of contacting with target cells. In fact, we have revealed that M-Sec plays a role in cell-to-cell transmission of the two structurally-related retroviruses, human immunodeficiency virus type 1 (HIV-1) and human T-cell leukemia virus type 1 (HTLV-1) [6–8].

Through an affinity-based chemical array screening, we previously discovered the small molecule compound NPD3064 [6], which bound M-Sec or M-Sec-containing molecular complex and inhibited the M-Sec-mediated TNT formation. We and others found that NPD3064 reduced not only the TNT formation but also HIV-1 production in monocyte-derived macrophages [6,9]. Consistent with this finding, the knockdown of M-Sec in the widely used HIV-1 target cell line U87 reduced the TNT formation and cell migration, and retarded the HIV-1 production in the multiple-round short-term infection systems [7].

We found that M-Sec also mediates an efficient infection of HTLV-1. When added to several co-culture assay systems, NPD3064 reduced the viral transmission to target CD4+ T cells [8]. The knockdown of M-Sec in HTLV-1-infected CD4+ T cells

reduced the viral transmission to target CD4[+] T cells in an *in vivo* model: when humanized mice were inoculated with the M-Sec knockdown HTLV-1-infected CD4[+] T cell line MT-2, the number of infected human lymphocytes in the liver, spleen, bone marrow, and peripheral blood was much lower than that of the parental MT-2-inoculated mice [8]. The M-Sec knock-down MT-2 showed weaker migratory activity and formed shorter TNTs than the parental cells [8].

The above-mentioned findings suggest that M-Sec contributes to an efficient cell-to-cell transmission of HIV-1 and HTLV-1 at least in part by promoting the motility/TNT formation of infected cells and their contact with target cells. To promote the TNT formation, M-Sec requires phosphatidylinositol 4,5-biphosphate, Ral (the small GTPase) and the exocyst complex (the downstream effector of Ral) [3,10], although the precise mechanism is unclear. Whether Ral and exocyst are requisite for M-Sec to promote cell motility is also unclear.

Interestingly, we recently found that M-Sec induces the accumulation of intracellular puncta of HTLV-1 Gag (the major structural protein essential for viral assembly) and the incorporation of Env (the envelope protein) into viral particles [11]. These results suggest that M-Sec contributes to the production of infectious particles of HTLV-1 because Gag and Env are critical for the assembly of viral particles and their attachment to target cells, respectively. However, the hypothesis has not been directly proven due to a technical difficulty: the infectivity of extracellular HTLV-1 is too low to detect in a quantitative manner. The molecular mechanism by which M-Sec affects the dynamics of HTLV-1 Gag and Env is not fully understood. Furthermore, it remains unexplored whether M-Sec has a similar effect on Gag and Env of HIV-1.

In this study, by taking advantage of the fact that the infectivity of extracellular HIV-1 is relatively easily detectable, we sought to clarify the effects of M-Sec on the dynamics of Gag and Env of HIV-1, and the resultant formation of infectious HIV-1 particles. The M-Sec-induced distribution alteration of viral Gag proteins is apparently unrelated to the M-Sec-induced formation of TNTs (the long extensions or protrusions of the plasma membrane). Therefore, we also sought to clarify whether these two different activities of M-Sec share the underlying molecular mechanism or not.

## Results

### M-Sec knockdown in U87 cells diminishes HIV-1 Gag puncta formation

To examine how the knockdown of M-Sec affects the intracellular distribution of HIV-1 Gag, we utilized the U87 cell system because the M-Sec knockdown in the cells retarded the HIV-1 production [7]. The stable M-Sec knockdown (ΔM-Sec) U87 cells were established (Fig 1A), and infected with HIV-1, together with the control cells (Fig 1B). The intracellular Gag puncta were easily visible in the control cells (Fig 1B, upper), but fewer in the ΔM-Sec cells (Fig 1B, lower). In the ΔM-Sec cells, most Gag proteins tended to distribute diffusely throughout the cytoplasm (Fig 1B, lower). When quantified, the average size of the Gag puncta (Fig 1C, upper, S1 Fig, upper) or the number of >0.25 µm Gag puncta (Fig 1C, lower, S1 Fig lower) in each Gag[+] ΔM-Sec cell was smaller than that in each Gag[+] control cell. Such change was rescued by the exogenous expression of mouse M-Sec in theΔM-Sec cells (S2 Fig), excluding the possibility of off-target effects. Similar results were obtained when monocyte-derived macrophages (MDMs) were infected with HIV-1 after the pretreatment with NPD3064 (S3A S3B Fig), which reduced the HIV-1 production in the cells [6,9]. The knockdown of M-Sec using siRNA was partial but detectable (S3C Fig), and the total number of Gag puncta in each Gag[+] M-Sec knockdown MDM was smaller than that in each Gag[+] control MDM (S3D Fig). These results indicate that M-Sec is involved in the puncta formation of HIV-1 Gag.

### Exogenous M-Sec expression in 293 cells induces the accumulation of HIV-1 Gag puncta

To further confirm the involvement of M-Sec in the puncta formation of HIV-1 Gag, we next performed the exogenous expression experiment using 293 cells, which hardly express endogenous M-Sec. The stable M-Sec-expressing 293 cells were established (Fig 2A), and transfected with the wild-type (WT) HIV-1 molecular clone, together with the parental cells (Fig 2B). The Gag puncta were detectable in the parental cells (Fig 2B, upper), but more visible in the M-Sec-expressing

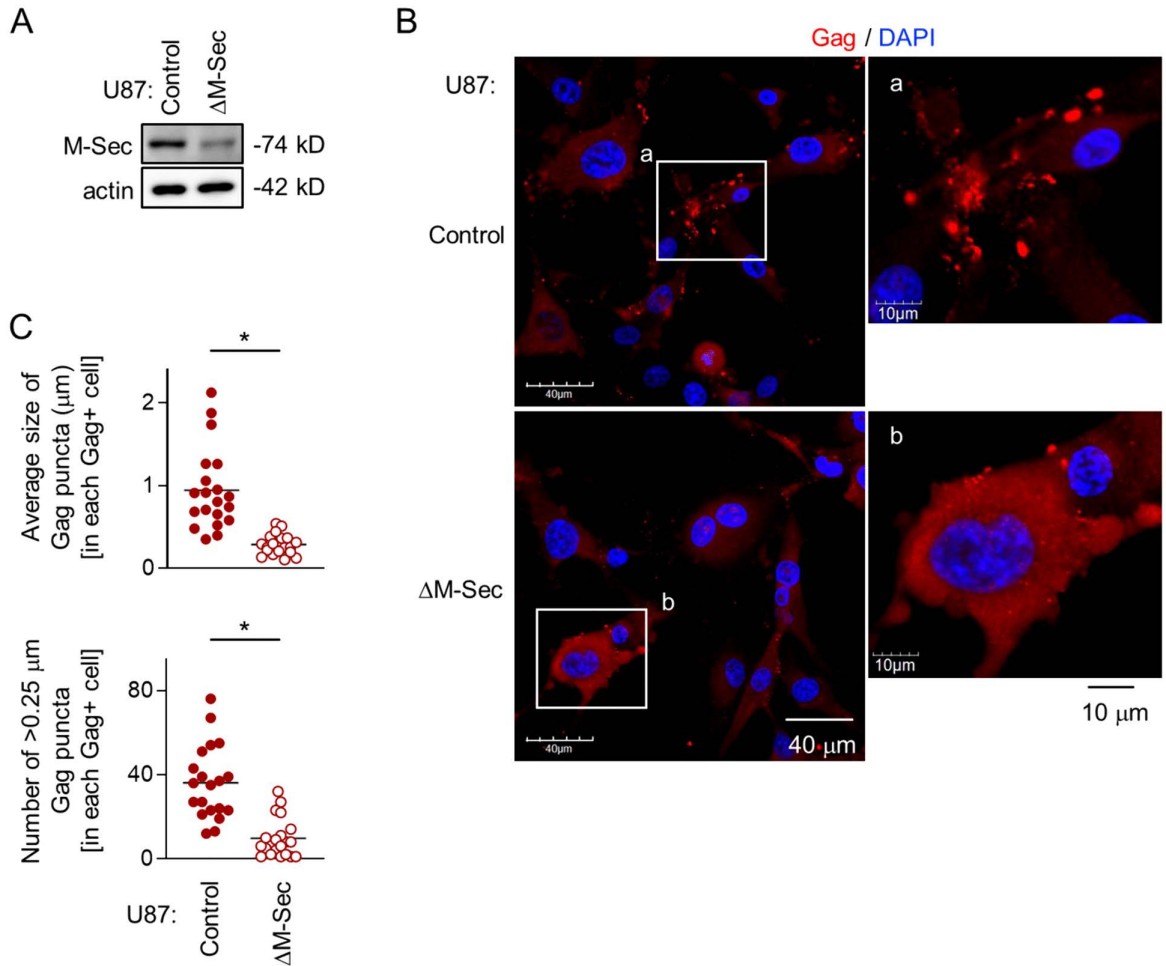

**Fig 1. Effect of M-Sec knockdown on Gag puncta formation in U87 cells. (A)** The control and stable M-Sec knockdown (ΔM-Sec) U87 cells were analyzed for their expression level of M-Sec by western blotting. β-actin blot is the loading control. **(B)** The control or ΔM-Sec U87 cells were infected with HIV-1, cultured for 2 days, and analyzed for Gag (red) by immunofluorescence. The nuclei were stained with DAPI (blue). In the right panels, the magnified images of "a" and "b" in the left panels are shown. Scale bars: 40 μm and 10 μm for the left and right panels, respectively. **(C)** The cells were analyzed as in **B**. In the upper panel, the average size of Gag puncta in each Gag+ cell is summarized (20 cells for each group). In the lower panel, the number of >0.25 μm Gag puncta in each Gag+ cell is summarized (20 cells for each group). The Gag signal larger than approximately 0.03 μm was considered puncta. *$p < 0.05$. Data shown are a representative of three independent experiments.

cells (Fig 2B, lower). When quantified, the size of the Gag puncta (Fig 2C, upper, S4 Fig upper) or the number of >0.25 μm Gag puncta (Fig 2C, lower, S4 Fig lower) in each Gag+ M-Sec-expressing cell was larger than that in each Gag+ parental cell. However, such difference was not observed for the defective Gag-expressing HIV-1 molecular clones, WMAA and G2A (Fig 2C). There was no obvious difference in the expression levels of those Gag proteins between the parental and M-Sec-expressing cells (Fig 2D). The WMAA harboring two mutations in the Gag CA domain (W184A/M185A) binds the inner leaflet of the plasma membrane inefficiently, and is defective in multimerization [12,13]. The G2A mutant lacking the N-terminal myristoylation signal in the Gag MA domain is defective in membrane binding [13]. Meanwhile, when assessed by the velocity sedimentation assay [14], the presence of M-Sec did not necessarily promote the multimerization of Gag (S5 Fig). These results suggest that multimerized Gag or Gag assembly intermediates get close to each other, which is detected as puncta, and that M-Sec does not affect the multimerization step *per se*, but induces

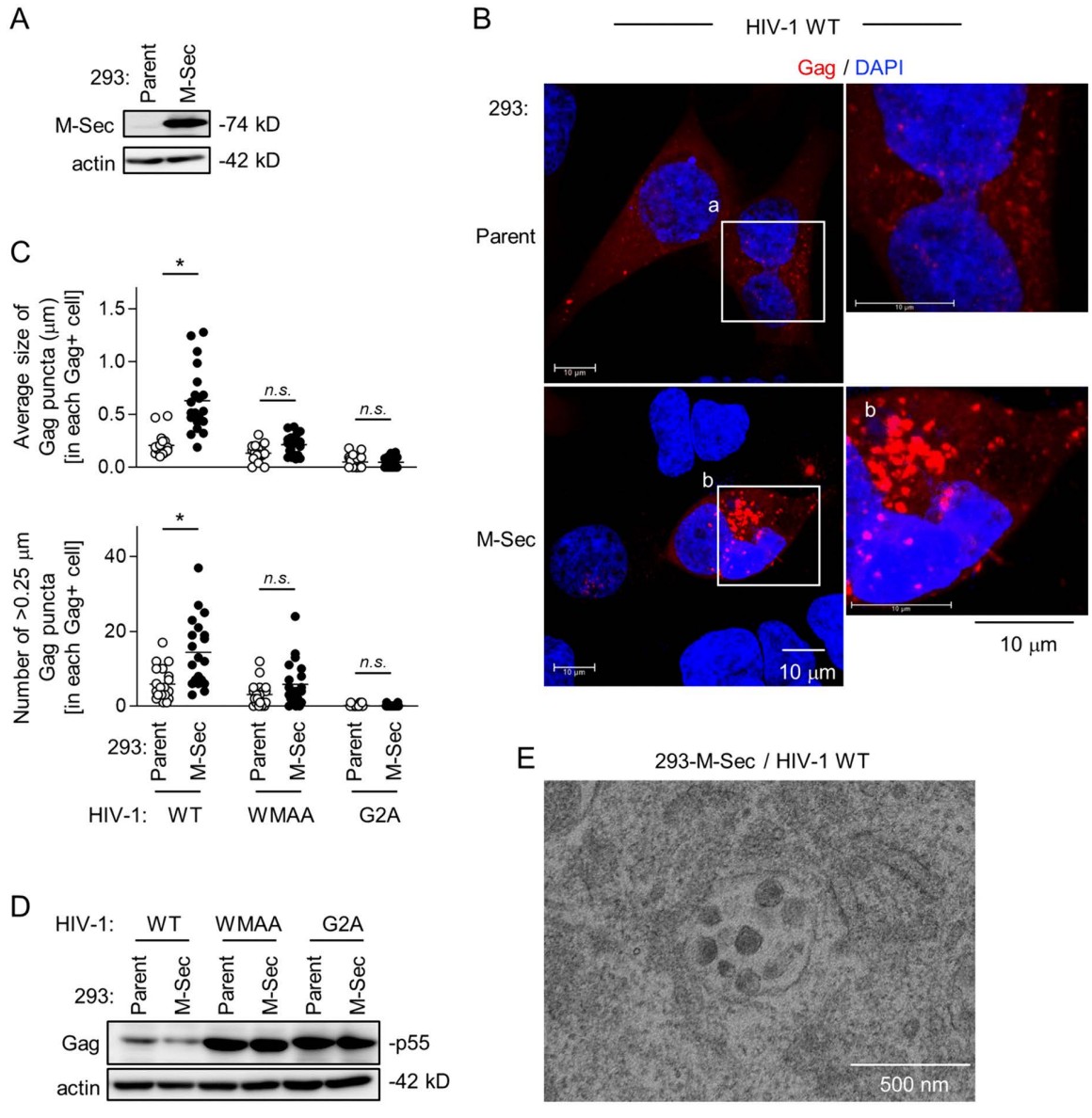

**Fig 2. Effect of M-Sec expression on Gag puncta formation in 293 cells. (A)** The parental and stable M-Sec-expressing 293 cells were analyzed for their expression level of M-Sec by western blotting. β-actin blot is the loading control. **(B)** The parental or M-Sec-expressing 293 cells were transfected with the wild-type (WT) HIV-1 molecular clone, cultured for 2 days, and analyzed for Gag (red) by immunofluorescence. The nuclei were stained with DAPI (blue). In the right panels, the magnified images of "a" and "b" in the left panels are shown. Scale bar: 10 μm. **(C)** The cells were analyzed as in **B**. In addition to WT, the WMAA and G2A mutants were used. In the upper panel, the average size of Gag puncta in each Gag$^+$ cell is summarized (20 cells for each group). In the lower panel, the number of >0.25 μm Gag puncta in each Gag$^+$ cell is summarized (20 cells for each group). The Gag signal larger than approximately 0.03 μm was considered puncta. *n.s.*, not significant. $^*p < 0.05$. Data shown are a representative of three independent experiments. **(D)** The parental or M-Sec-expressing 293 cells were transfected with the indicated HIV-1 molecular clone, cultured for 2 days, and analyzed for their expression level of Gag (p55) by western blotting. β-actin blot is the loading control. **(E)** The M-Sec-expressing 293 cells were transfected with the WT HIV-1 molecular clone, cultured for 2 days, and analyzed for virus-like particles using an electron microscope. Scale bar: 500 nm.

the accumulation of puncta. Consistent with this idea, the accumulation of virus-like particles was often detected in the M-Sec-expressing cells by an electron microscopic analysis (Fig 2E).

## M-Sec induces the accumulation of HIV-1 Gag puncta independently of other viral components

Even when transfected with the GFP-fused Gag expression plasmid in place of the HIV-1 molecular clone, the M-Sec-expressing 293 cells formed larger Gag puncta than the parental cells (Fig 3A). The size of the Gag puncta (Fig 3B, upper, S6 Fig upper) or the number of >0.25 µm Gag puncta (Fig 3B, lower, S6 Fig lower) in each Gag⁺ M-Sec-expressing cell was larger than that in each Gag⁺ parental cell, which was the case not only for the immunofluorescence using fixed cells (Fig 3B, left, S6 Fig) but also the live cell imaging (Fig 3B, right, S6 Fig). There was no obvious difference in the expression levels of Gag-GFP proteins between the parental and M-Sec-expressing cells (Fig 3C). These results indicate that M-Sec induces the accumulation of HIV-1 Gag independently of other viral proteins. The effect of M-Sec on Gag might not be nonspecific, since there was no obvious distribution alteration in Nef (S7 Fig), the HIV-1 pathogenic protein that has the myristoylation signal as with Gag.

## M-Sec promotes the co-localization of Env with accumulated Gag puncta

To examine whether M-Sec is also involved in the regulation of Env, we first compared the control and the M-Sec knockdown (ΔM-Sec) U87 cells. When infected with HIV-1, the control cells formed the large Env-positive compartments, the signal of which overlapped with the accumulated Gag puncta (Fig 4A, upper). In contrast, the ΔM-Sec cells hardly formed such large Env-positive compartment (Fig 4A, lower), and showed weaker Gag/Env co-localization than the control cells (Fig 4B, S8 Fig).

To further test whether M-Sec is indirectly involved in the regulation of Env, we next compared the parental and M-Sec-expressing 293 cells. When transfected with the HIV-1 molecular clone, the M-Sec-expressing cells showed stronger Gag/Env co-localization than the parental cells (Fig 5A, 5B, S9 Fig upper) and the increased local density of Env (Fig 5A, 5C, S9 Fig lower). These results were well consistent with those of experiments using control/M-Sec knockdown U87 cells (see Fig 4). When transfected with the Env expression plasmid alone in place of the HIV-1 molecular clone, there was no obvious difference in the density of Env distribution in Env-positive compartments (S10 Fig, "Env alone"). In contrast, when co-transfected with the expression plasmids of Env and Gag, the M-Sec-expressing cells showed an increased local density of Env than the parental cells (S10 Fig, "Env + Gag"). Collectively, these results suggest that M-Sec induces the accumulation of Gag puncta, which leads to the enhanced Gag/Env co-localization and the formation of large or dense Env-positive compartments.

## M-Sec promotes the production of infectious viral particles

To examine how the observed effect of M-Sec on Gag and Env results in the production of viral particles and their infectivity, we first compared the control and the M-Sec knockdown (ΔM-Sec) U87 cells. The HIV-1 Env is synthesized as the precursor form (gp160), which is cleaved into mature forms (gp120 and gp41). When infected with HIV-1, the control and ΔM-Sec cells showed similar expression levels of those Env proteins (Fig 6A). However, the virus-like particles produced by the ΔM-Sec cells contained a lesser amount of gp120 and gp41 than those by the control cells (Fig 6B), even after the normalization to mature Gag (p24) in the virus-like particles (Fig 6C). The processing of Gag was almost unchanged (S11 Fig). The ΔM-Sec cells produced slightly lesser amount of virus than the control because the reverse transcriptase (RT) activity in the supernatants of the ΔM-Sec cells was slightly weaker than that of the control cells (Fig 6D), which was confirmed by p24 Gag ELISA (S12 Fig). Importantly, when assessed using TZM-bl cells as target cells [15], viruses produced by the ΔM-Sec cells showed weaker infectivity than those by the control cells, even after the normalization to the RT activity (Fig 6E). The weaker infectivity was rescued by the exogenous expression of mouse M-Sec in the ΔM-Sec cells (S13 Fig).

To further confirm that M-Sec is involved in the regulation of viral production and infectivity, we next compared the parental and M-Sec-expressing 293 cells. When transfected with the HIV-1 molecular clone, the parental and M-Sec-expressing cells

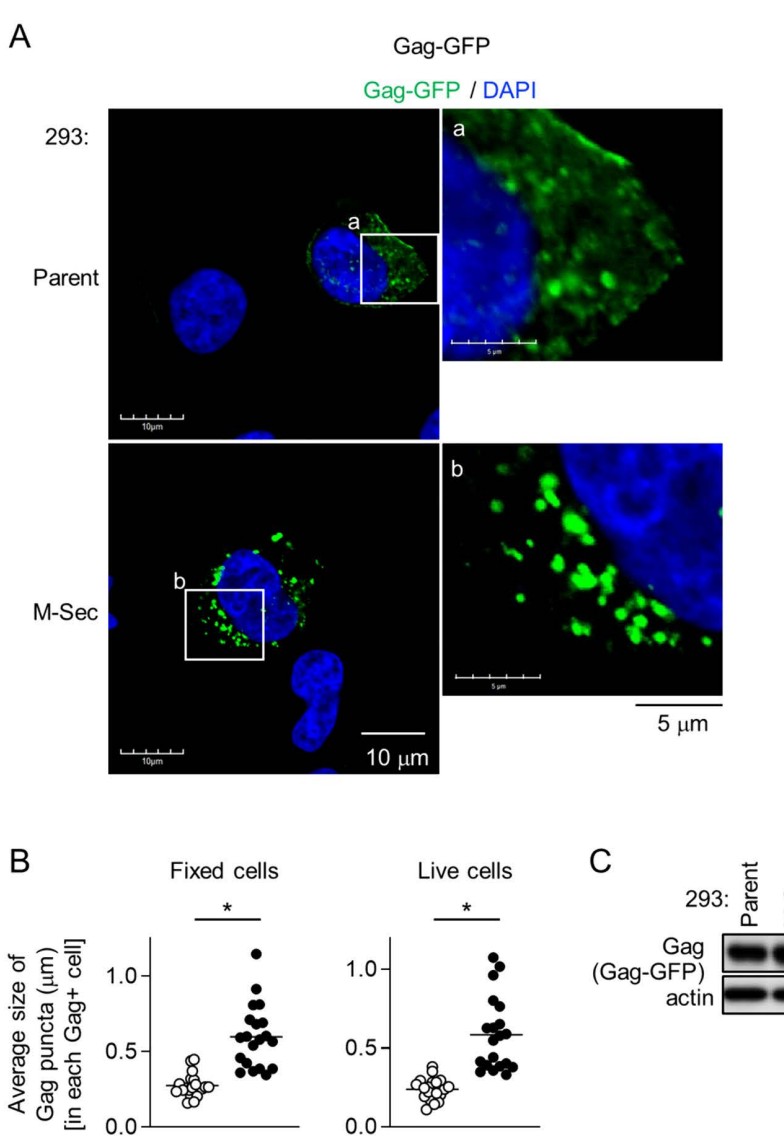

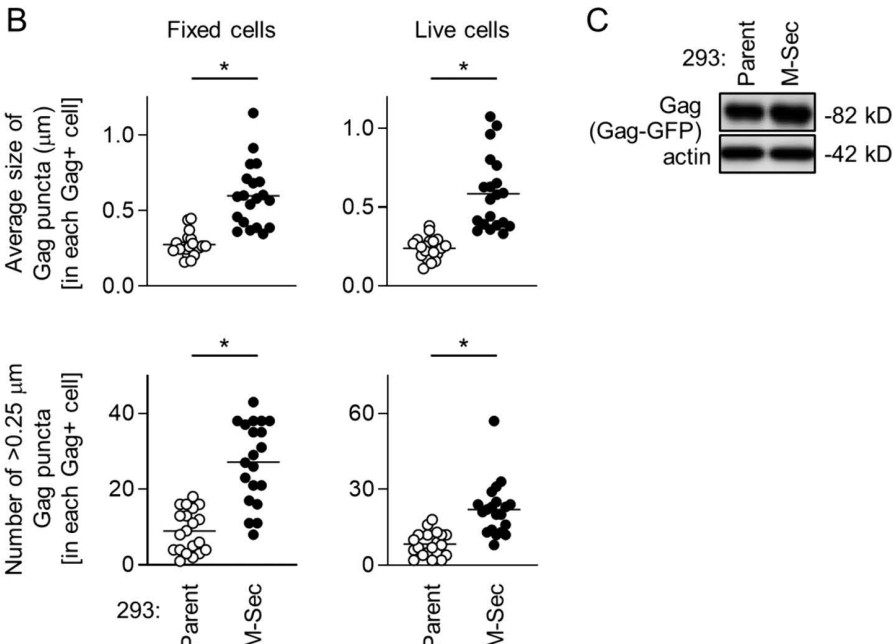

**Fig 3. Effect of M-Sec on Gag puncta formation in the absence of other viral components. (A)** The parental or stable M-Sec-expressing 293 cells were transfected with the Gag-GFP expression plasmid, cultured for 24 hours, and analyzed for Gag-GFP (green) by immunofluorescence. The nuclei were stained with DAPI (blue). In the right panels, the magnified images of "a" and "b" in the left panels are shown. Scale bars: 10 μm and 5 μm for the left and right panels, respectively. **(B)** The cells were analyzed as in **A** (left panels). In the right panels, the transfected cells were analyzed by live cell imaging. In the upper panel, the average size of Gag puncta in each Gag⁺ cell is summarized (20 cells for each group). In the lower panel, the number

of >0.25 μm Gag puncta in each Gag⁺ cell is summarized (20 cells for each group). The Gag signal larger than approximately 0.03 μm was considered puncta. *$p < 0.05$. Data shown are a representative of three independent experiments. **(C)** The parental or M-Sec-expressing 293 cells were transfected with the Gag-GFP expression plasmid, cultured for 24 hours, and analyzed for their expression level of Gag-GFP by western blotting using anti-Gag antibody. β-actin blot is the loading control.

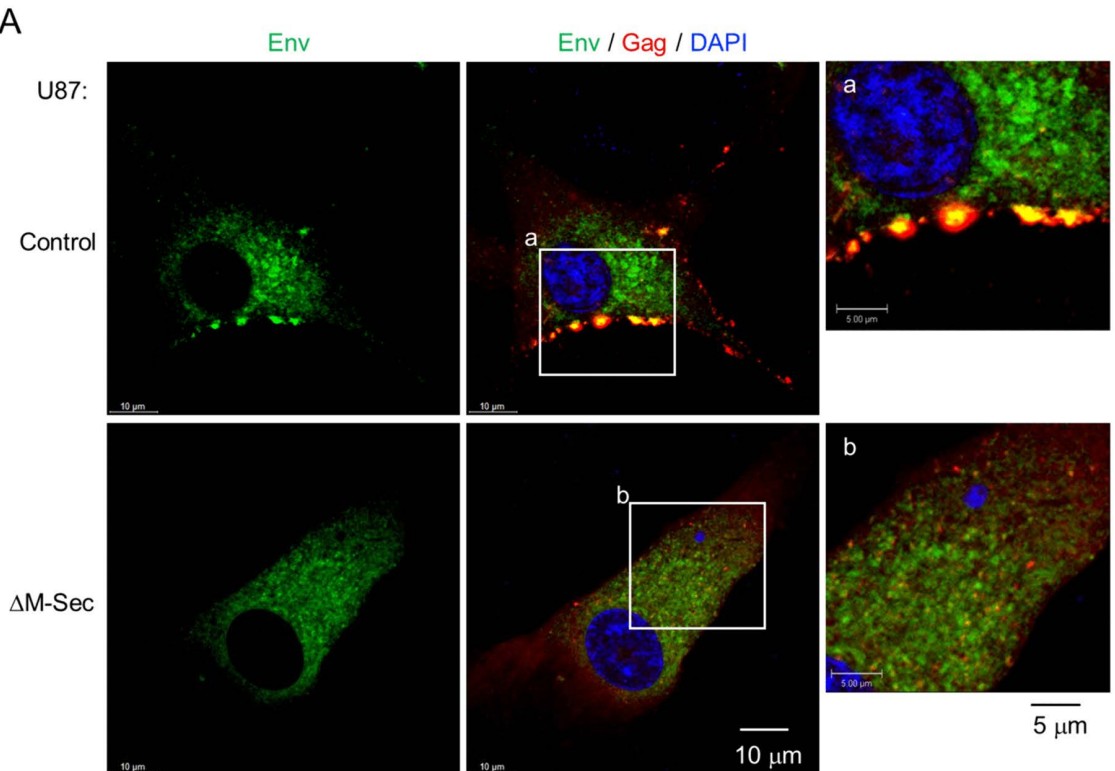

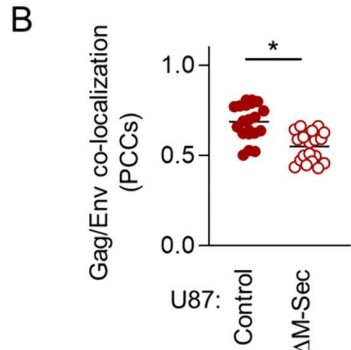

**Fig 4. Effect of M-Sec knockdown on Env distribution and Gag/Env co-localization in U87 cells. (A)** The control or stable M-Sec knockdown (ΔM-Sec) U87 cells were infected with HIV-1, cultured for 2 days, and analyzed for Env (green) and Gag (red) by immunofluorescence. The nuclei were stained with DAPI (blue). In the right panels, the magnified images of "a" and "b" in the middle panels are shown. Scale bars: 10 μm and 5 μm for the left/middle and right panels, respectively. **(B)** The cells were analyzed as in **A**. The co-localization of Gag and Env was quantified as Pearson's correlation coefficients (PCCs) (20 cells for each group). *$p < 0.05$. Data shown are a representative of three independent experiments.

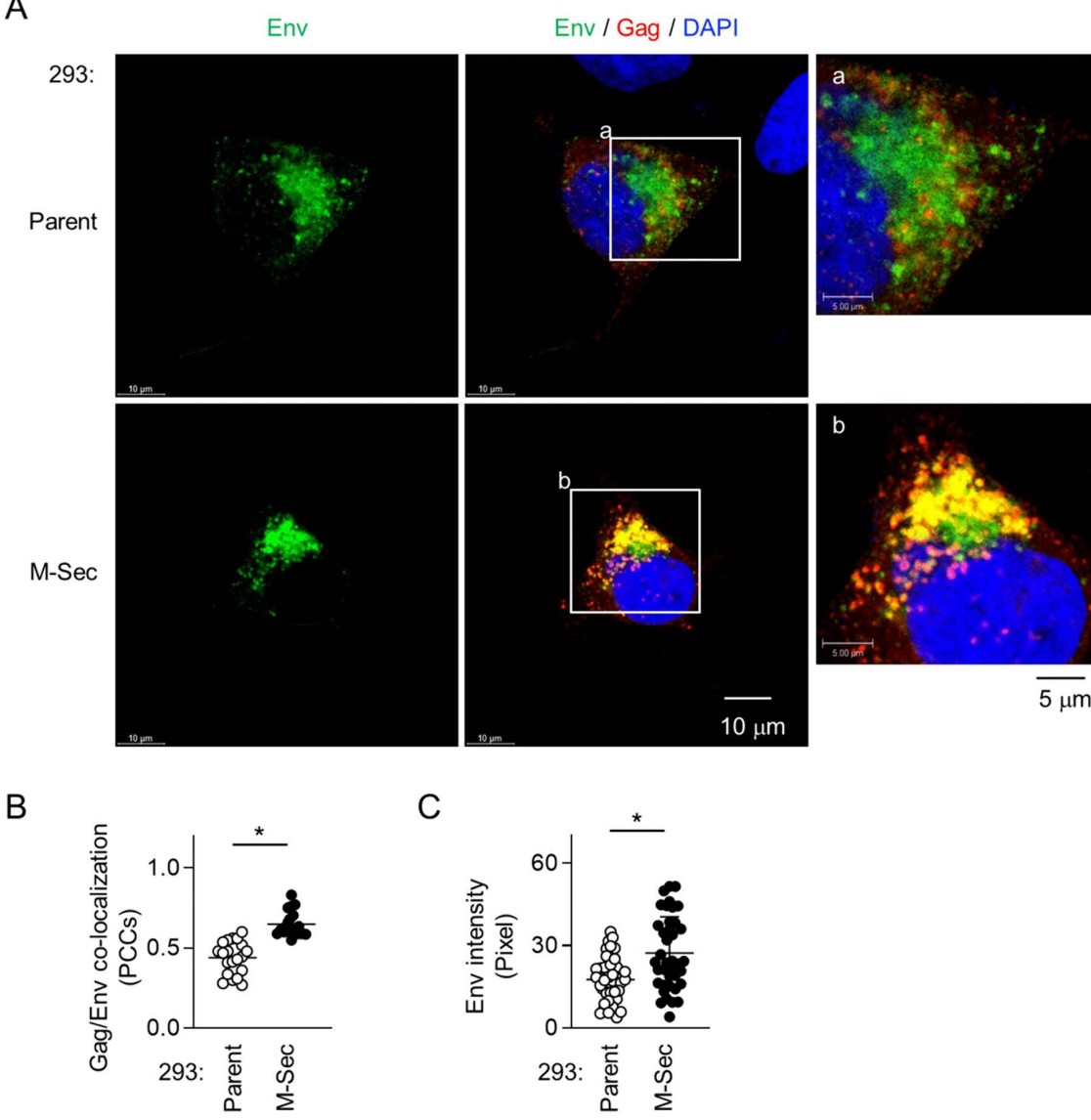

**Fig 5. Effect of M-Sec expression on Gag/Env co-localization in 293 cells. (A)** The parental or stable M-Sec-expressing 293 cells were transfected with the wild-type HIV-1 molecular clone, cultured for 2 days, and analyzed for Env (green) and Gag (red) by immunofluorescence. The nuclei were stained with DAPI (blue). In the right panels, the magnified images of "a" and "b" in the middle panels are shown. Scale bars: 10 μm and 5 μm for the left/middle and right panels, respectively. **(B, C)** The cells were analyzed as in **A**. In **B**, the co-localization of Gag and Env was quantified as Pearson's correlation coefficients (PCCs) (20 cells for each group). In **C**, the density of Env was quantified by randomly selecting Env-positive areas (90 μm$^2$ for each area, two areas for each cell, 20 cells for each group). $^*p < 0.05$. Data shown are a representative of three independent experiments.

showed a similar expression level of gp160 (Fig 7A). However, the virus-like particles produced by the M-Sec-expressing cells contained higher amount of gp120 and gp41 than those by the parental cells (Fig 7B), even after the normalization to p24 Gag in the virus-like particles (Fig 7C). The processing of Gag was almost unchanged (S14 Fig). The M-Sec-expressing cells produced slightly higher amount of virus than the parental cells (Fig 7D, S15 Fig), and those viruses showed stronger infectivity than those by the parental cells (Fig 7E). Because there was no detectable difference in the size and weight

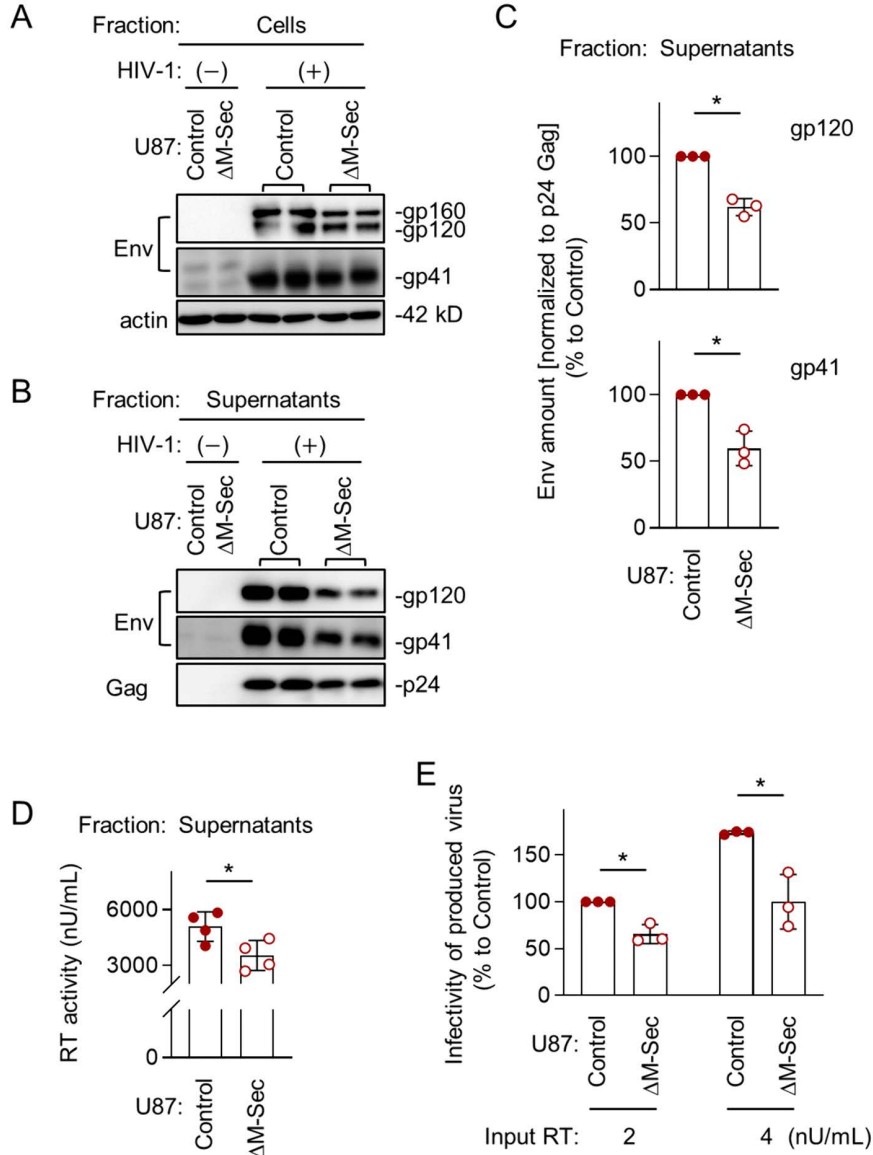

**Fig 6. Effect of M-Sec knockdown on Env incorporation into virus-like particles and viral infectivity in U87 cells. (A)** The control or stable M-Sec knockdown (ΔM-Sec) U87 cells were left uninfected or infected with HIV-1, cultured for 2 days, and analyzed for their expression level of Env (gp160 and gp120 in the top blot, and gp41 in the middle blot) by western blotting. β-actin blot is the loading control. **(B)** The control or ΔM-Sec U87 cells were left uninfected or infected with HIV-1, and cultured for 2 days. The virus-like particles in the supernatants were collected by centrifugation, and analyzed for their amount of gp120 (top), gp41 (middle), and p24 Gag (bottom) by western blotting. **(C)** The virus-like particles were analyzed as in **B**. The density of gp120- or gp41 band was normalized to that of p24 Gag band, and represented as a percentage relative to that of the control cells (n = 3). $^*p < 0.05$. **(D, E)** The control or ΔM-Sec U87 cells were infected with HIV-1, and cultured for 2 days. In **D**, the supernatants were collected, and analyzed for the activity of reverse transcriptase (RT) by qPCR (n = 4). In **E**, the supernatants were collected, and analyzed for viral infectivity using TZM-bl cells as the target cells. Two different inputs were tested (2 and 4 nU/mL of RT activity). The infectivity is represented as a percentage relative to that of the control cells with the input of 2 nU/mL (n = 3).

(contrast) of virus-like particles themselves between the parental and M-Sec-expressing cells by mass photometry (S16 Fig), these results suggest that M-Sec increases viral infectivity by promoting the incorporation of Env into viral particles.

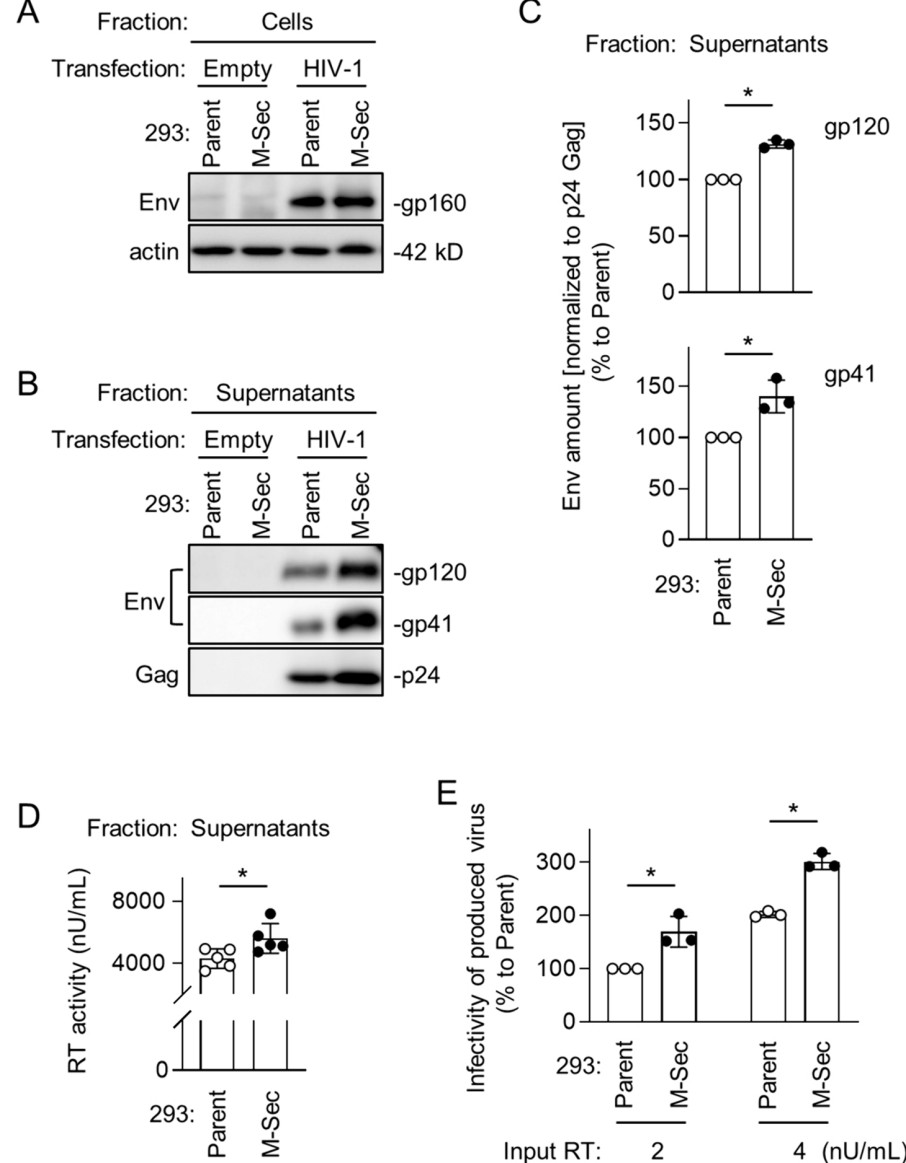

**Fig 7. Effect of M-Sec expression on Env incorporation into virus-like particles and viral infectivity in 293 cells. (A)** The parental or stable M-Sec-expressing 293 cells were transfected with the empty plasmid or HIV-1 molecular clone, cultured for 2 days, and analyzed for their expression level of gp160 by western blotting. β-actin blot is the loading control. **(B)** The parental or M-Sec-expressing 293 cells were transfected with the empty plasmid or HIV-1 molecular clone, and cultured for 2 days. The virus-like particles in the supernatants were collected by centrifugation, and analyzed for their amount of gp120 (top), gp41 (middle), and p24 Gag (bottom) by western blotting. **(C)** The cells were analyzed as in **B**. The density of gp120- or gp41 band was normalized to that of p24 Gag band, and represented as a relative to that of the parental cells (n = 3). $^{*}p < 0.05$. **(D, E)** The parental or M-Sec-expressing 293 cells were transfected with the HIV-1 molecular clone, and cultured for 2 days. In **D**, the supernatants were collected, and analyzed for the activity of reverse transcriptase (RT) by qPCR (n = 5). In **E**, the supernatants were collected, and analyzed for viral infectivity using TZM-bl cells as the target cells. Two different inputs were tested (2 and 4 nU/mL of RT activity). The infectivity is represented as a percentage relative to that of the parental cells with the input of 2 nU/mL (n = 3).

## M-Sec-mediated accumulation of Gag puncta depends on phosphatidylinositol 4,5-biphosphate

To promote the TNT formation, M-Sec requires phosphatidylinositol 4,5-biphosphate (PIP2) [3,10], although the precise mechanism is unclear. M-Sec binds PIP2 through its N-terminal lysine-rich motifs [10]. Thus, we next examined whether the M-Sec-mediated accumulation of Gag puncta, a likely key initial step to the production of infectious viral particles, also depends on PIP2, which is the most abundant phosphoinositide and usually concentrated in the plasma membrane [16–18].

Consistent with the M-Sec - PIP2 binding [10], the parental and M-Sec-expressing 293 cells showed a different PIP2 distribution (S17 Fig, left) [11]: when engineered to express GFP-fused pleckstrin homology domain of phospholipase Cδ (GFP-PLCδ-PH), the probe to visualize PIP2 in living cells [19,20], the M-Sec-expressing cells, but not the parental cells, contained large GFP-PLCδ-PH-positive structures. Such difference was not observed when we used the probe GFP-Akt-PH [20] to visualize another phosphoinositide (phosphatidylinositol 3,4,5-triphosphate; PIP3) (S17 Fig, right). These results suggest that the M-Sec binding alters the intracellular distribution of PIP2, but not PIP3.

To confirm the requirement of PIP2 for the M-Sec-mediated accumulation of Gag puncta, we depleted PIP2 in the M-Sec-expressing 293 cells by the over-expression of inositol polyphosphate 5-phosphatase type IV (5ptaseIV) [21,22]. The wild-type (WT), 5ptaseIV, but not the defective mutant lacking the 5-phosphatase domain (Δ1), reduced the average size of the Gag puncta (S18 Fig, upper) or the number of >0.25 μm Gag puncta (S18 Fig, lower) in each Gag+ M-Sec-expressing cell. Consistent with this result, M-Sec failed to induce the accumulation of puncta when we used the 6A2T Gag mutant (S19 Fig), which is defective in PIP2 binding [23].

## M-Sec-mediated accumulation of Gag puncta depends on Ral and the exocyst complex

To promote the TNT formation, M-Sec also requires the small GTPase Ral and the exocyst complex [3,10], although the precise mechanism is not fully understood. M-Sec binds an active Ral through its C-terminal positively charged domains [10], and Ral binds the exocyst complex [24,25], the octameric protein complex involved in vesicle trafficking of cell surface or secreted proteins [26,27]. Thus, we examined whether the M-Sec-mediated accumulation of Gag puncta also depends on Ral and the exocyst complex.

BQU57 is the widely used inhibitor of Ral [28,29]. Endosidin2 (ES2) is the inhibitor of EXOC7 (a.k.a Exo70), a subunit of the exocyst complex [30,31]. When pretreated with either BQU57 or ES2, and infected with HIV-1, the control U87 cells showed the M-Sec knockdown cell-like phenotypes, i.e., the smaller Gag puncta (Fig 8A, 8B, S20 Fig top and middle) and the weaker Gag/Env co-localization (Fig 8A, 8C, S20 Fig bottom). Similar results were obtained for MDMs (S21 Fig). Moreover, viruses produced by MDMs pretreated with BQU57 or ES2 showed weaker infectivity than those produced by the DMSO-treated control MDMs (Fig 8D).

The exocyst complex is composed of eight subunits EXOC1 to EXOC8 [26,27]. To test whether the exocyst complex is the effector required for the stimulatory effect of M-Sec on HIV-1, we generated the EXOC2- or EXOC3 knockdown (ΔEXOC2 or ΔEXOC3) M-Sec-expressing 293 cells (Fig 9A). We could not obtain the M-Sec-expressing 293 cells that stably showed a reduced level of EXOC1, EXOC6, or EXOC7. The M-Sec expression was unaffected by the knockdown of EXOC2 or EXOC3 (Fig 9A). However, the knockdown of EXOC2 or EXOC3 hindered the accumulation of Gag puncta originally observed in the M-Sec-expressing cells (Fig 9B). When quantified, there was no difference in the average size of the Gag puncta (Fig 9C, S22 Fig upper), the Gag/Env co-localization (Fig 9D, S22 Fig lower), and the infectivity of produced virus-like particles (Fig 9E) between the parental cells and M-Sec-expressing EXOCs knockdown cells. There was no further reduction in the Gag puncta formation and Gag/Env co-localization in the EXOC3 knockdown parental cells (S23 Fig). These results suggest that the exocyst complex is required for M-Sec to promote the production of infectious HIV-1 particles.

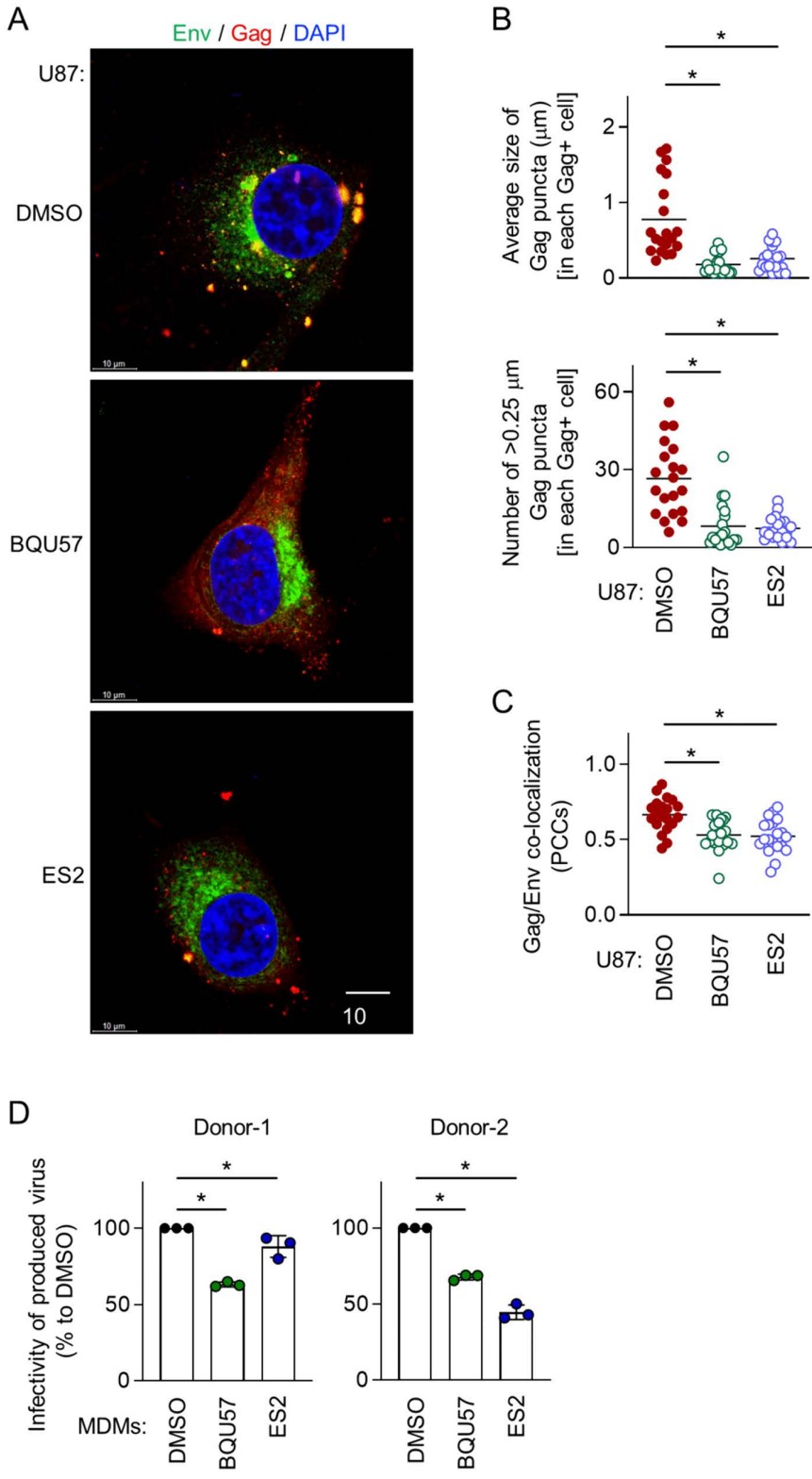

**Fig 8. Effect of inhibition of Ral or the exocyst complex on Gag puncta formation and Gag/Env co-localization in U87 cells, or viral infectivity in peripheral blood monocyte-derived macrophages. (A - C)** The control U87 cells were pretreated with DMSO, 10 μM BQU57 or 10 μM ES2 for 2

days, and infected with HIV-1. Then, the cells were cultured for 2 days in the presence of DMSO, 10 µM BQU57 or 10 µM ES2, and analyzed for Env (green) and Gag (red) by immunofluorescence. The nuclei were stained with DAPI (blue). Scale bar: 10 µm. In the upper panel of **B**, the average size of Gag puncta in each Gag⁺ cell is summarized (20 cells for each group). In the lower panel of **B**, the number of >0.25 µm Gag puncta in each Gag⁺ cell is summarized (20 cells for each group). The Gag signal larger than approximately 0.03 µm was considered puncta. In **C**, the co-localization of Gag and Env was quantified as Pearson's correlation coefficients (PCCs) (20 cells for each group). $^*p < 0.05$. Data shown are a representative of three independent experiments. **(D)** Peripheral blood monocyte-derived macrophages (MDMs) obtained from two different donors (Donor-1 and Donor-2) were pretreated with DMSO, 10 µM BQU57 or 10 µM ES2 for 2 days, and infected with HIV-1. Then, the cells were cultured for 2 days in the presence of DMSO, 10 µM BQU57 or 10 µM ES2. The supernatants were collected, and analyzed for viral infectivity using TZM-bl cells as the target cells (the viral input: 4 nU/mL reverse transcriptase activity). The infectivity is represented as a percentage relative to that of the DMSO-treated control MDMs (n = 3).

## Discussion

We previously demonstrated that M-Sec mediates the efficient transmission of HIV-1, and proposed that the possible underlying mechanism is its stimulatory effect on cell motility and TNT formation [6,7]. In addition to this indirect effect on cellular behaviors, the present study revealed a direct effect on the late steps of the life cycle of HIV-1: M-Sec affects the intracellular distribution of HIV-1 Gag proteins and thereby promotes the production of infectious viral particles in a manner dependent on PIP2 and the Ral - the exocyst complex cascade. Thus, it appears that both the indirect and direct effects enable M-Sec to contribute to the efficient HIV-1 transmission.

M-Sec binds PIP2 through its N-terminal basic region [10]. The present study showed that M-Sec also induces the redistribution of PIP2, but not of PIP3 (S17 Fig). Such membrane ruffling-associated PIP2 redistribution or increased local density of PIP2 may cause the accumulation of HIV-1 Gag puncta because the Gag protein binds PIP2 through its N-terminal polybasic region [32–34] and the PIP2 depletion reduced the accumulation of Gag puncta (S18 Fig). To increase the local density of PIP2, M-Sec may also utilize the Ral - the exocyst complex cascade because M-Sec itself diffusely localizes throughout the cytoplasm [35]. Consistent with this idea, the inhibition of Ral or EXOC7 (Fig 8) or the knockdown of EXOC2 or EXOC3 hindered the M-Sec-mediated accumulation of Gag puncta (Fig 9). Thus, it is likely that the molecular components by which M-Sec promotes the formation of TNTs and the accumulation of HIV-1 Gag puncta are largely overlapping, although it remains unclear how those apparently unrelated phenomena are governed by these overlapping components.

It is well known that the incorporation of Env into viral particles is essential for the production of infectious progeny virus. Although the precise mechanism of HIV-1 Env incorporation into viral particles is not currently understood, a likely mechanism is the incorporation through an indirect or direct interaction with Gag [36]. The present study also suggests the Env-Gag interaction because Env shows the increased local density and co-localization with Gag when the Gag puncta accumulate in the presence of M-Sec (Fig 4, Fig 5), but not when Gag is absent even in the presence of M-Sec (S10 Fig). Thus, it is possible that the accumulated Gag puncta recruit Env through the indirect or direct interaction, which promotes the incorporation of Env and the production of infectious viral particles (Figs 6, 7). However, whether the enhanced Env incorporation is the sole cause of the enhanced infectivity remains unexplored. It will be necessary to test whether M-Sec also promotes the incorporation of cellular proteins into viral particles or the formation of PIP2-enriched viral particles.

We recently reported a similar effect of M-Sec on Gag and Env of another human retrovirus, HTLV-1 [11]. In the HTLV-1 study, we demonstrated the importance of PIP2 [11]. However, the involvement of Ral and the exocyst complex remained unexplored. How M-Sec affects the production of infectious HTLV-1 viral particles also remained unexplored because the infectivity of extracellular HTLV-1 is too low to detect in a quantitative manner. In fact, the cell-free infection of HTLV-1 is quite inefficient [37]. Thus, the present study revealed a previously unreported mechanism and activity by which M-Sec contributes to the efficient transmission of these retroviruses.

The exocyst complex is involved in vesicle trafficking of cell surface or secreted proteins, such as the tethering of secretory vesicles to the plasma membrane, and composed of eight different proteins EXOC1 to EXOC8 [26,27]. Interestingly,

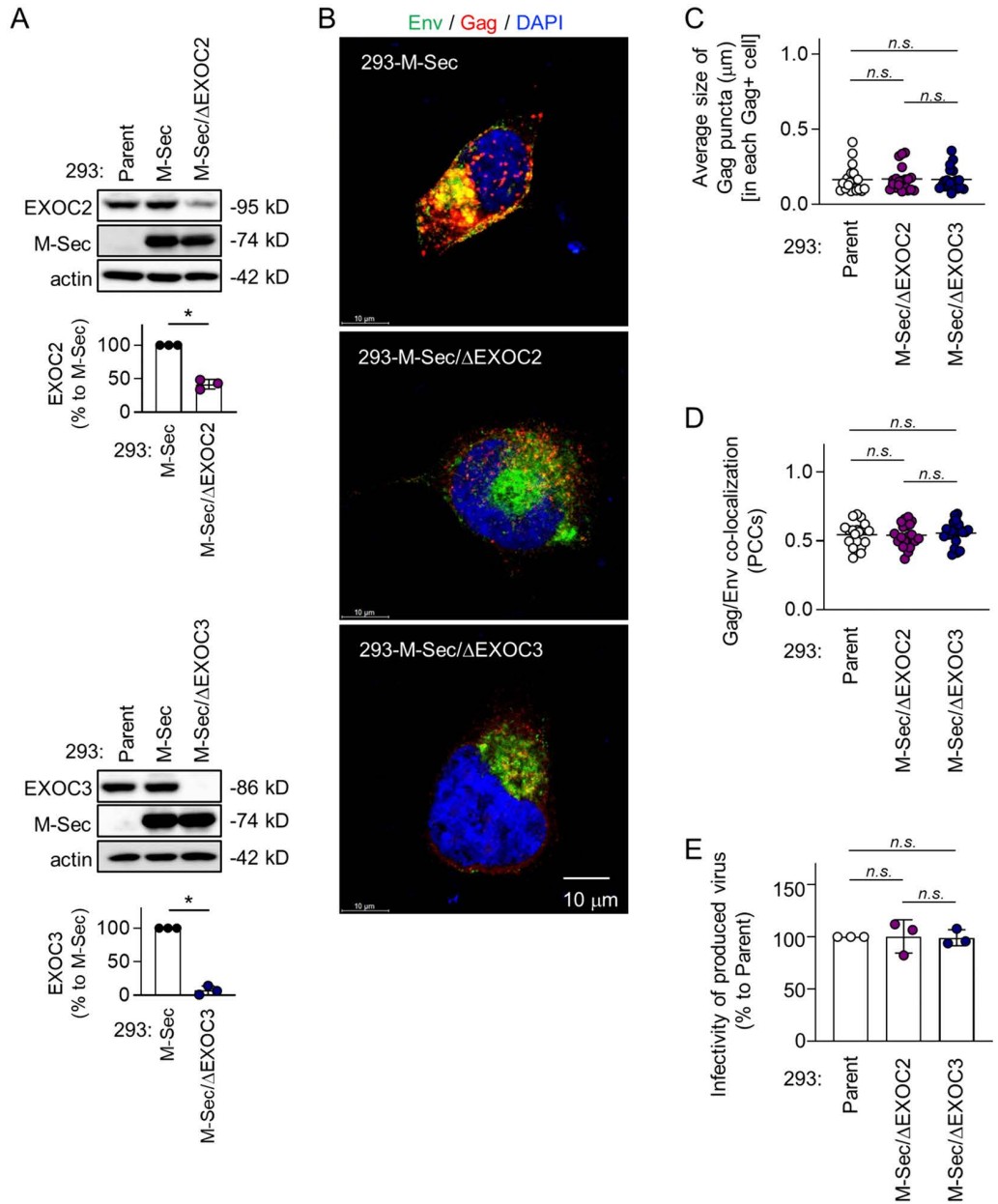

**Fig 9. Effect of knockdown of EXOC2 or EXOC3 on Gag puncta formation, Gag/Env co-localization, and viral infectivity in M-Sec-expressing 293 cells. (A)** In the upper blots, the parental (Parent), M-Sec-expressing (M-Sec), or stable EXOC2 knockdown M-Sec-expressing (M-Sec/ΔEXOC2) 293 cells were analyzed for their expression of EXOC2 or M-Sec by western blotting. β-actin blot is the loading control. In the graph, the density of EXOC2 band of the M-Sec/ΔEXOC2 293 cells is represented as a percentage relative to that of the M-Sec 293 cells (n = 3). In the lower blots, the stable EXOC3 knockdown M-Sec-expressing (M-Sec/ΔEXOC3) 293 cells were analyzed for their expression of EXOC3. In the graph, the density of EXOC3 band of the M-Sec/ΔEXOC3 293 cells is represented as a percentage relative to that of the M-Sec 293 cells (n = 3). *$p < 0.05$. **(B)** The M-Sec, M-Sec/ΔEXOC2, or M-Sec/ΔEXOC3 293 cells were transfected with the wild-type HIV-1 molecular clone, cultured for 2 days, and analyzed for Env (green) and Gag (red) by immunofluorescence. The nuclei were stained with DAPI (blue). Scale bar: 10 μm. **(C, D)** The parental (Parent), M-Sec/ΔEXOC2, or M-Sec/ΔEXOC3 293 cells were transfected and analyzed as in **B**. In **C**, the average size of Gag puncta in each Gag+ cell is summarized (20 cells for each group). In **D**, the co-localization of Gag and Env was quantified as Pearson's correlation coefficients (PCCs) (20 cells for each group). *n.s.*, not significant. Data shown are a representative of three independent experiments. **(E)** The parental (Parent), M-Sec/ΔEXOC2, or M-Sec/ΔEXOC3 293 cells were transfected with the wild-type HIV-1 molecular clone, and cultured for 2 days. The supernatants were collected, and analyzed for viral infectivity using TZM-bl cells as the target cells (the input: 4 nU/mL of reverse transcriptase (RT) activity). The infectivity is represented as a percentage relative to that of the parental cells (n = 3).

recent studies revealed the involvement of several EXOC proteins in viral infection. For example, the knockdown of EXOC2 (a.k.a Sec5) inhibited the infection of SARS-CoV-2 [38]. The knockdown of EXOC7 (a.k.a Exo70) inhibited the egression/secretion of dengue virus [39]. The over-expression of EXOC5 (a.k.a Sec10) stimulated the infection of HSV-1 and VSV [40]. These results and the present study suggest an important role of the exocyst complex in the infection of various viruses. Among HIV-1 target cells, M-Sec is highly expressed in macrophages, but not in CD4$^+$ T cells [3]. Thus, M-Sec can contribute to the transmission of HIV-1 from macrophages, but not from CD4$^+$ T cells. Meanwhile, EXOC proteins are evolutionary conserved and ubiquitously expressed [26,27]. Thus, it will be intriguing to test how EXOC proteins contribute to HIV-1 infection from CD4$^+$ T cells.

We previously reported that the M-Sec knockdown reduced the production of the wild-type virus, but not that of Nef-deficient virus [7]. The Nef-deficient virus replicated less efficiently than the wild-type virus [7], and Nef was required for the M-Sec-mediated tunneling nanotube formation [7]. Thus, we speculated that Nef played a role in the M-Sec-mediated viral production through the tunneling nanotube formation. However, the speculation might not necessarily fully explain the reduced production of the wild-type virus by the M-Sec knockdown because the M-Sec-mediated Gag puncta formation was independent of Nef. Thus, M-Sec may contribute to HIV-1 production in both Nef-dependent and Nef-independent manners.

The difference in viral infectivity induced by the M-Sec knockdown or over-expression correlated with the difference in viral production (Figs 6, 7). However, its degree was subtle. Thus, to clarify its impact on a long-term viral replication, further study is necessary.

The accumulation of Gag puncta is likely the first key step for the M-Sec-mediated production of infectious HIV-1 particles. PIP2 is present not only in the inner leaflet of the plasma membrane but also at multiple intracellular vesicles [16]. In addition to M-Sec and Gag, the two subunits of the exocyst complex, EXOC1 (a.k.a Sec3) and EXOC7 bind PIP2 [41,42]. The Ral - the exocyst complex cascade is involved in vesicular trafficking [43–45]. When combined with these findings, the present study may raise the possibility that the binding of M-Sec to Ral potentiates the vesicle trafficking activity of the exocyst complex, which relates to the accumulation of viral particles at the intracellular vesicles. Consistent with this idea, the accumulation of virus-like particles was often detected in the M-Sec-expressing cells by an electron microscopic analysis (Fig 2E), the structures of which resemble the virus-containing compartments (VCCs) in macrophages [46]. In fact, the signal of Env in MDMs and the M-Sec-expressing 293 cells overlapped with the signal of CD81 (S24 Fig), a widely-used marker of VCCs [46]. Further study is necessary to prove the hypothesis that M-Sec plays a role in the formation of VCCs. Despite the unresolved questions, the present study identifies the additional and previously unreported mechanism by which M-Sec promotes the transmission of HIV-1.

## Materials and methods

### Ethics statement

Peripheral blood was collected from healthy donors after written informed consent had been obtained in accordance with the Declaration of Helsinki. This study was approved by the Kumamoto University medical ethics committee.

### U87 cells and stable M-Sec knockdown

The U87 (U-87MG) glioma cell line was obtained through the American Type Culture Collection (ATCC) and maintained with DMEM-10% FCS. The stable M-Sec knockdown U87 cells were established using the MISSION short hairpin (sh) RNA lentiviral transduction system (Sigma), as described previously [8]. pLKO.1-puro-CMV-tGFP vector that expresses shRNA that targets human M-Sec (TRCN0000330220) or a scrambled nontargeting control (both from Sigma) was used. The control or M-Sec knockdown cells were selected by culturing with 0.2 μg/mL puromycin. The expression level of tGFP and M-Sec in these cells was verified by flow cytometry using Cytek Northern Lights (Cytek Biosciences) and western blotting, respectively. In a selected experiment, U87 cells were transfected with a mouse M-Sec expression plasmid [10]

using Lipofectamine 3000 (Invitrogen) [11]. The control or M-Sec knockdown U87 cells stably expressing mouse M-Sec were also established by culturing with 300 μg/mL G418.

### Human monocyte-derived macrophages (MDMs)

Human MDMs were prepared as described previously [6]. To facilitate the adherence of monocytes to culture dishes, mononuclear cells were suspended in RPMI 1640 medium containing a low concentration of FCS (1%). The adherent monocytes were differentiated into macrophages by culturing with RPMI 1640-10% FCS containing 100 ng/ml recombinant human M-CSF (a gift from Morinaga Milk Industry, Kanagawa, Japan) for 5–7 days. In a selected experiment, the knockdown of M-Sec was performed using siRNA (Dharmacon) and RNAiMAX (Invitrogen), as described previously [47]. siRNAs used were non-targeting siRNA (pool #2; D-001206–14) and M-Sec-specific siRNA (#4; D-012267–17) [7].

### Recombinant HIV-1 infection of U87 cells or MDMs

In the previous studies, to assess the effect of M-Sec on the cell-to-cell transmission of HIV-1, we employed the usual infection system using U87 cells and MDMs as the target cells [6,7]. Throughout this study, to exclude an effect by multiple infections and simply assess the effect of M-Sec on the late life cycle of HIV-1 in initially-infected cells, we employed the single-round infection systems that allow the production of mature progeny virus [48]. Specifically, we used vesicular stomatitis virus envelope G glycoprotein (VSV-G)-pseudotyped JRFL strain and NL43 strain for U87 cells and MDMs, respectively. To assess the infectivity of produced virus, the wild-type (=HIV-1 Env-expressing) JRFL or NL43 strain was used. To prepare those viruses, 293 cells were co-transfected with each HIV-1 molecular clone and the VSV-G plasmid at a ratio of 4:1 [48]. At 2 days post-transfection, the supernatants were collected as the viral stock and stored at -70°C until use. U87 cells and MDMs were infected with those viruses (input Gag concentration: 100 ng/mL) for 2 hours at 37°C [48], and further cultured for subsequent analyses.

### Stable M-Sec over-expression in 293 cells

The 293A cell line was purchased from Invitrogen and maintained with DMEM-10% FCS. The 293 cells stably over-expressing M-Sec were established using the self-inactivating lentiviral transduction system (TaKaRa-Bio), as described previously [11]. The pLVSIN-EF1α-Hyg vector (TaKaRa-Bio) carrying the N-terminally Flag-tagged human M-Sec DNA was used. The transduced cells were selected by culturing with 200 μg/mL hygromycin. The expression level of M-Sec in the cells was verified by immunofluorescence and western blotting.

### Stable knockdown of EXOC2 or EXOC3 in M-Sec-expressing 293 cells

The stable knockdown of EXOC2 or EXOC3 in the parental or M-Sec-expressing 293 cells was performed using the MISSION short hairpin (sh)RNA lentiviral transduction system (Sigma) [8]. pLKO.1-puro-CMV-tGFP vector that expresses shRNA that targets human EXOC2 (TRCN0000116103) or EXOC3 (TRCN0000074304) (both from Sigma) was used. The transduced cells were selected by culturing with 1 μg/mL puromycin. The expression level of EXOC2 or EXOC3 in the cells, together with that of M-Sec, was verified by western blotting.

### HIV-1 molecular clone transfection into 293 cells

To assess the effect of M-Sec on the late life cycle of HIV-1, we also transfected HIV-1 molecular clones into 293 cells. The parental or genetically-modified 293 cells were seeded onto 12-well plates ($1.8 \times 10^5$ cells/well), cultured for 1 day, and transfected with 1 μg plasmid using 3 μL Lipofectamine 3000 (Invitrogen) [11]. After 6 hours of transfection, the culture medium was replaced with fresh medium, and the cells were further cultured for subsequent analyses. The molecular clones used were the wild-type JRFL strain, or the wild-type NL43 strain and its mutants WMAA, G2A, or 6A2T [12,13,23].

In selected experiments, we also used the plasmid expressing the codon-optimized Gag-GFP fusion protein (SynGag-GFP) [49]. As a reference, the plasmid expressing Env (pCXN-FLenv) [50] or Nef-GFP [51].

### Chemical inhibitors

In this study, NPD3064, BQU57, and ES2 were used. NPD3064 inhibits M-Sec-mediated formation of TNTs [6], and was synthesized at Sundia MedTech (Shanghai, China). BQU57 and ES2 were purchased from Sigma. BQU57 is thought to bind a GDP-bound form of Ral (RalA or RalB) and thereby inhibit its interaction with effector molecules, including EXOC2 (a.k.a Sec5) and EXOC8 (a.k.a Exo84) [28,29]. ES2 is thought to bind the C-terminal domain of EXOC7 and thereby inhibit the plasma membrane localization of EXOC7 and the function of the exocyst complex [30,31]. These inhibitors were dissolved in DMSO and added to the medium at a final concentration of 10 μM (0.1% v/v). The same volume of DMSO was added as a vehicle control.

### Western blotting

Western blotting was performed as described previously [11]. The cells or the virus-like particles in the culture super-natants were lysed with Nonidet P-40 lysis buffer containing protease/phosphatase inhibitors, and subjected to western blotting. The antibodies used were as follows: anti-M-Sec (F-6; Santa Cruz Biotechnology), anti-p24 Gag (#65–004; BioAcademia), anti-gp120 Env (KD247; provided by S. Matsushita, Kumamoto University, Japan), anti-gp41 Env (2F5; obtained through BEI Resources, NIAID, NIH), anti-EXOC2 (#12751–1-AP; Proteintech), anti-EXOC3 (#14703–1-AP; Proteintech), anti-GFP (B-2; Santa Cruz Biotechnology), and anti-β-actin (EPR16769; Abcam). Detection was performed using HRP-labeled secondary antibodies (Cytiva or Proteintech), western blot ultra-sensitive HRP substrate (TaKaRa-Bio), and ImageQuant LAS4000 image analyzer (Cytiva).

### Velocity sedimentation

In a selected experiment, the velocity sedimentation assay was performed to monitor the multimerization of Gag [14]. The parental or stable M-Sec-expressing 293 cells were transfected with the wild-type HIV-1 molecular clone (the NL43 strain), and cultured for 2 days. The cell lysates were prepared and ultracentrifuged through sucrose gradients. Fractions were collected from centrifuge tubes (from top to bottom), and analyzed for p55 Gag by western blotting.

### Virus-like particles, western blotting, mass photometry

The virus-like particles in the culture supernatants were collected by centrifugation at 20,000 x g for 60 minutes at 4°C [11]. The pellets were dissolved in SDS-PAGE sample buffer and subjected to western blotting to quantify the amounts of Gag and Env in the virus-like particles. Alternatively, the pellets were resuspended in 2% glutaraldehyde and subjected to the analysis using the mass photometry (KaritroMP; Refeyn) to measure the size and weight (contrast) of the virus-like particles.

### Immunofluorescence

Immunofluorescence was performed as described previously [11]. The cells were fixed with paraformaldehyde, permeabilized with Triton X-100, and stained with anti-p24 Gag (24–2; Santa Cruz Biotechnology) or anti-gp120 Env (2G12; Syd Labs) followed by AlexaFluor633-labeled anti-mouse IgG or AlexaFluor568-labeled anti-human IgG (both from Molecular Probes). In a selected experiment, anti-CD81 (5A6; BioLegend) was used. DAPI (Molecular Probes) was used to visualize nuclei. Signals were visualized using confocal laser-scanning microscope FV1200 (Olympus) or Leica TCS SP8 (Leica). Image processing was performed using FV10 Viewer ver. 4.2 (Olympus) or LAS X Office software (Leica). The size of Gag puncta was measured by ImageJ 1.52n software. The Gag signal larger than approximately 0.03 μm was considered

puncta. The Env signal was also quantified by the ImageJ software. The Gag/Env co-localization was calculated using the JACoP plugin in the ImageJ software program and represented by Pearson's correlation coefficients.

### PIP2 depletion and immunofluorescence

The plasmid expressing the wild-type or defective Δ1 mutant of 5ptaseIV (pcDNA4TO/Myc5ptaseIV-WT or pcDNA4TO/Myc5ptaseIV-Δ1) has been used to deplete cellular PIP2 [22]. To analyze the effect of PIP2 depletion on Gag, the M-Sec-expressing 293 cells were co-transfected with the 5ptaseIV plasmid and SynGag-GFP plasmid, and analyzed for Gag by immunofluorescence.

### Live cell imaging

The plasmid expressing GFP-PLCδ-PH and GFP-Akt-PH have been used to monitor cellular distribution of PIP2 and PIP3, respectively [19,20]. To visualize the effect of M-Sec expression on the distribution of PIP2 or PIP3 in living cells, the parental or M-Sec-expressing 293 cells were transfected with the plasmid expressing GFP-PLCδ-PH or GFP-Akt-PH (both from Addgene), and analyzed by live cell imaging using the confocal laser-scanning microscope TCS SP8.

### Transmission electron microscopy

Transmission electron microscopic analysis was performed as described previously [52]. The M-Sec-expressing 293 cells transfected with the HIV-1 molecular clone were fixed with 2% glutaraldehyde and 1% osmium tetroxide (both from TAAB Laboratories Equipment), and dehydrated with ethanol. The samples were embedded in Epon812 resin (TAAB Laboratories Equipment), and ultrathin sections on copper grids (Nisshin-EM) were stained with Mayer's hematoxylin solution and lead citrate. The stained samples were observed and recorded using Hitachi 7600 transmission electron microscope (Hitachi High-Technologies) at 80 kV.

### Reverse transcriptase activity and p24 Gag ELISA

The viral production was monitored by measuring the reverse transcriptase activity in the culture supernatants by the real-time PCR [53]. The viruses in the supernatants were lysed and added to the mixture containing the MS2 RNA template (Merck). Real-time PCR was performed using TB Green Premix Ex Taq II (TaKaRa-Bio) and LightCycler (Roche). The primers used are as follows: 5'- TCCTGCTCAACTTCCTGTCGAG-3' (forward) and 5'- CACAGGTCAAACCTCC TAGGAATG-3' (reverse). The viral production was also monitored by p24 Gag ELISA (Rimco, Okinawa, Japan).

### Viral infectivity

The viral infectivity was assessed using TZM-bl cells (NIH AIDS Research & Reference Program) as described previously [15]. The cells were seeded, incubated with the diluted viruses normalized to the reverse transcriptase activity, and cultured for 48 hours. The viral infectivity was assessed by measuring the HIV-1 Tat-mediated induction of β-galactosidase activity in the target cells using a β-Galactosidase Enzyme Assay System (Promega). The absorbance of the wells was measured at 420 nm using a microplate reader.

### Statistical analysis

Differences between two groups were analyzed by unpaired Student's $t$-test or Kolmogorov-Smirnov test. Differences between three or more groups were analyzed by one-way ANOVA or two-way ANOVA. Pearson correlation analysis was also performed. All statistical analyses were conducted using PRISM 10 (GraphPad). $P$ values < 0.05 were considered significant.

## Supporting information

**S1 Fig. Additional experimental sets for Fig 1C.**
(PDF)

**S2 Fig. Effect of exogenous expression of mouse M-Sec on Gag puncta formation in M-Sec knockdown U87 cells.**
(**A**, **B**) The control or stable M-Sec knockdown (ΔM-Sec) U87 cells were transfected with the empty vector ("−") or the GFP-fused mouse M-Sec (GFP-mM-Sec) expression plasmid ("+"), and cultured for 3 days. In **A**, the cells were analyzed for their expression level of GFP-mM-Sec by western blotting. β-actin blot is the loading control. In **B**, the cells were infected with HIV-1, cultured for 2 days, and analyzed for Gag by immunofluorescence. The number of >0.25 μm Gag puncta in each Gag⁺ cell is summarized (20 cells for each group). *n.s.*, not significant. *$p < 0.05$.
(PDF)

**S3 Fig. Effect of NPD3064 or M-Sec knockdown on Gag puncta formation in peripheral blood monocyte-derived macrophages.** (**A**) Peripheral blood monocyte-derived macrophages (MDMs) were pretreated with DMSO or 10 μM NPD3064 for 2 days, and infected with HIV-1. After two or four days of infection (dpi), the cells were analyzed for Gag (red) by immunofluorescence. The nuclei were stained with DAPI (blue). In the right panels, the magnified images of "a" and "b" in the middle panels are shown. Scale bars: 10 μm and 5 μm for the left/middle and right panels, respectively. (**B**) MDMs were infected for 4 days, and analyzed as in **A**. In the upper panel, the average size of Gag puncta in each Gag⁺ cell is summarized (20 cells for each group). In the lower panel, the number of >0.25 μm Gag puncta in each Gag⁺ cell is summarized (20 cells for each group). The Gag signal larger than approximately 0.03 μm was considered puncta. *$p < 0.05$. Data shown are a representative of three independent experiments. (**C**, **D**) In **C**, MDMs were transfected with the control siRNA or M-Sec-specific siRNA, cultured for 4 days, and analyzed for their expression level of M-Sec by western blotting. β-actin blot is the loading control. In **D**, MDMs were transfected with the control siRNA or M-Sec-specific siRNA, and cultured for 2 days. Then, the cells were infected with HIV-1, cultured for 2 days, and analyzed for Gag by immunofluorescence. The total number of Gag puncta (>0.03 μm) in each Gag⁺ cell is summarized (20 cells for each group). Data shown are a representative of three independent experiments.
(PDF)

**S4 Fig. Additional experimental sets for Fig 2C.**
(PDF)

**S5 Fig. Effect of M-Sec expression on Gag multimerization in 293 cells.** The parental or stable M-Sec-expressing 293 cells were transfected with the wild-type (WT) HIV-1 molecular clone, and cultured for 2 days. The cell lysates were prepared and ultracentrifuged through sucrose gradients. Fractions were collected from centrifuge tubes (from top to bottom), and analyzed for Gag (p55) by western blotting. Sedimentation coefficients (S values) are also shown. 10S, 80S/150S, and 500S/750S contain nascent-, oligomerized-, and multimerized Gag, respectively [14].
(PDF)

**S6 Fig. Additional experimental sets for Fig 3B.**
(PDF)

**S7 Fig. Effect of M-Sec expression on Nef distribution in 293 cells.** The parental or stable M-Sec-expressing 293 cells were transfected with the Nef-GFP expression plasmid, cultured for 2 days, and analyzed for Nef-GFP (green) by immunofluorescence. The nuclei were stained with DAPI (blue). Scale bar: 10 μm.
(PDF)

**S8 Fig. Additional experimental sets for Fig 4B.**
(PDF)

**S9 Fig. Additional experimental sets for Fig 5B and 5C.**
(PDF)

**S10 Fig. Effect of M-Sec expression on Env distribution in 293 cells.** The parental or stable M-Sec-expressing 293 cells were transfected with the Env expression plasmid (Env alone), or co-transfected with the expression plasmids of Env and Gag (Env + Gag). Then, the cells were cultured for 2 days, and analyzed for Env by immunofluorescence. The density of Env was quantified by randomly selecting Env-positive areas (90 μm$^2$ for each area, two areas for each cell, total 20 areas for each group). *n.s.*, not significant. $^*p < 0.05$. Data shown are a representative of three independent experiments.
(PDF)

**S11 Fig. Effect of M-Sec knockdown on Gag processing in U87 cells.** (**A**, **B**) The control or stable M-Sec knockdown (ΔM-Sec) U87 cells were left uninfected or infected with HIV-1, and cultured for 2 days. The virus-like particles in the supernatants were collected by centrifugation, and analyzed for their amount of p55 Gag, p41 Gag, and p24 Gag by western blotting. In **B**, the percentage of each Gag protein to the total Gag proteins is summarized (n = 3).
(PDF)

**S12 Fig. Effect of M-Sec knockdown on viral production in U87 cells.** The control or stable M-Sec knockdown (ΔM-Sec) U87 cells were infected with HIV-1, and cultured for 2 days. The supernatants were collected, and analyzed for the activity of reverse transcriptase (RT) by qPCR (n = 3, left), or p24 Gag by ELISA (n = 3, right). $^*p < 0.05$.
(PDF)

**S13 Fig. Effect of exogenous expression of mouse M-Sec on viral infectivity in M-Sec knockdown U87 cells.** The stable M-Sec knockdown U87 cells (ΔM-Sec), the ΔM-Sec U87 cells stably expressing mouse M-Sec (ΔM-Sec/mM-Sec), or the control U87 cells stably expressing mM-Sec (Control/mM-Sec) were infected with HIV-1, and cultured for 2 days. The supernatants were collected, and analyzed for viral infectivity using TZM-bl cells as the target cells (the viral input: 2 nU/mL reverse transcriptase activity). The infectivity is represented as a percentage relative to that of the ΔM-Sec U87 cells (n = 3). *n.s.*, not significant. $^*p < 0.05$.
(PDF)

**S14 Fig. Effect of M-Sec expression on Gag processing in 293 cells.** (**A**, **B**) The parental or stable M-Sec-expressing 293 cells were transfected with the empty plasmid or HIV-1 molecular clone, and cultured for 2 days. The virus-like particles in the supernatants were collected by centrifugation, and analyzed for their amount of p55 Gag, p41 Gag, and p24 Gag by western blotting. In **B**, the percentage of each Gag protein to the total Gag proteins is summarized (n = 3).
(PDF)

**S15 Fig. Effect of M-Sec expression on viral production in 293 cells.** The parental or stable M-Sec-expressing 293 cells were transfected with the empty plasmid or HIV-1 molecular clone, and cultured for 2 days. The supernatants were collected, and analyzed for the activity of reverse transcriptase (RT) by qPCR (n = 3, left), or p24 Gag by ELISA (n = 3, right). $^*p < 0.05$.
(PDF)

**S16 Fig. Effect of M-Sec expression on the size and weight of virus-like particles in 293 cells.** (**A**, **B**) The parental or stable M-Sec-expressing 293 cells were transfected with the empty plasmid or HIV-1 molecular clone, and cultured for 2 days. The virus-like particles in the supernatants were collected by centrifugation, and analyzed for their size and weight (contrast) by mass photometry. In **A**, a typical example is shown. In **B**, the results of three independent assays were summarized. a.u., arbitrary unit. *n.s.*, not significant.
(PDF)

PLOS Pathogens

**S17 Fig. Effect of M-Sec expression on cellular distribution of PIP2 or PIP3 in 293 cells.** The parental or stable M-Sec-expressing 293 cells were transfected with the plasmid expressing either GFP-PLCδ-PH (PIP2 probe) or GFP-Akt-PH (PIP3 probe), cultured for 24 hours, and analyzed for the GFP signal by live cell imaging. Scale bar: 10 μm. (PDF)

**S18 Fig. Effect of PIP2 depletion on M-Sec-mediated accumulation of Gag puncta in 293 cells.** The M-Sec-expressing 293 cells were co-transfected (1:1) with the Gag-GFP expression plasmid and the empty plasmid (−), or 5ptaseIV plasmid (the wild-type (WT) or defective Δ1 mutant). The cells were cultured for 2 days and analyzed for Gag-GFP by immunofluorescence. In the upper panel, the average size of Gag puncta in each Gag+ cell is summarized (20 cells for each group). In the lower panel, the number of >0.25 μm Gag puncta in each Gag+ cell is summarized (20 cells for each group). The Gag signal larger than approximately 0.03 μm was considered puncta. *n.s.*, not significant. *$p < 0.05$. (PDF)

**S19 Fig. Effect of M-Sec expression on puncta formation of mutant Gag defective in PIP2-binding in 293 cells.** (**A**) The parental or stable M-Sec-expressing 293 cells were transfected with the wild-type (WT) HIV-1 molecular clone or 6A2T mutant that is defective in PIP2 binding due to mutations in the N-terminal highly basic region in the Gag MA domain. Then, the cells were cultured for 2 days, and analyzed for Gag signal by immunofluorescence. In the upper panel, the average size of Gag puncta in each Gag+ cell is summarized (20 cells for each group). In the lower panel, the number of >0.25 μm Gag puncta in each Gag+ cell is summarized (20 cells for each group). The Gag signal larger than approximately 0.03 μm was considered puncta. *n.s.*, not significant. *$p < 0.05$. (**B**) The parental or stable M-Sec-expressing 293 cells were transfected with the indicated HIV-1 molecular clone, cultured for 2 days, and analyzed for their expression level of Gag (p55) by western blotting. β-actin blot is the loading control. (PDF)

**S20 Fig. Additional experimental sets for Fig 8B and 8C.** (PDF)

**S21 Fig. Effect of inhibition of Ral or the exocyst complex on Gag puncta formation and Gag/Env co-localization in peripheral blood monocyte-derived macrophages.** (**A**, **B**) Peripheral blood monocyte-derived macrophages (MDMs) were pretreated with DMSO, 10 μM BQU57 or 10 μM ES2 for 2 days, and infected with HIV-1. Then, the cells were cultured for 2 days in the presence of DMSO, 10 μM BQU57 or 10 μM ES2, and analyzed for Gag and Env by immunofluorescence. In **A**, the average size of Gag puncta or the number of >0.25 μm Gag puncta in each Gag+ cell is summarized (20 cells for each group). In **B**, the co-localization of Gag and Env was quantified as Pearson's correlation coefficients (PCCs) (20 cells for each group). *$p < 0.05$. Data shown are a representative of three independent experiments. (PDF)

**S22 Fig. Additional experimental sets for Fig 9C and 9D.** (PDF)

**S23 Fig. Effect of EXOC3 knockdown on Gag puncta formation and Gag/Env co-localization in parental 293 cells.** (**A**) The parental 293 cells (Parent) or the EXOC3 knockdown parental 293 cells (Parent/ΔEXOC3) were analyzed for their expression of EXOC3 by western blotting. β- actin is the loading control. (**B**, **C**) The parental 293 cells (Parent) or the EXOC3 knockdown parental 293 cells (Parent/ΔEXOC3) were transfected with the wild-type HIV-1 molecular clone, cultured for 2 days, and analyzed for Env and Gag by immunofluorescence. In the upper panel of **B**, the average size of Gag puncta in each Gag+ cell is summarized (20 cells per group). In the lower panel of **B**, the number of >0.25 μm Gag puncta in each Gag+ cell is summarized (20 cells per group). The Gag signal larger than approximately 0.03 μm was considered

puncta. In **C**, the co-localization of Gag and Env was quantified as Pearson's correlation coefficients (PCCs) (20 cells per group). *n.s.*, not significant.
(PDF)

**S24 Fig. Co-localization of Env and CD81 in peripheral blood monocyte-derived macrophages and M-Sec-expressing 293 cells.** (**A**) Peripheral blood monocyte-derived macrophages (MDMs) were left uninfected or infected with HIV-1, cultured for 2 days, and analyzed for Env (green) and CD81 (magenta) by immunofluorescence. The nuclei were stained with DAPI (blue). Scale bar: 10 μm. (**B**) The parental or stable M-Sec-expressing 293 cells were transfected with the wild-type HIV-1 molecular clone, cultured for 2 days, and analyzed as in **A**.
(PDF)

**S1 Raw images. Original images for blots.**
(ZIP)

## Acknowledgments

We thank K. Nasu and I. Suzu for the technical and secretarial assistance, respectively. The following reagent was obtained through BEI Resources, NIAID, NIH: HIV-1 gp41 2F5 (ARP-1475).

## Author contributions

**Conceptualization:** Shinya Suzu.

**Investigation:** Reem M. Mahmoud, Kazuaki Monde, Nami Monde, Takaaki Koma, Hidenobu Mizuno.

**Resources:** Masateru Hiyoshi, Randa A. Abdelnaser, Kazuaki Monde, Takaaki Koma, Sara A. Habash, Naofumi Takahashi, Yosuke Maeda, Akira Ono.

**Writing – original draft:** Reem M. Mahmoud, Shinya Suzu.

**Writing – review & editing:** Masateru Hiyoshi, Randa A. Abdelnaser, Kazuaki Monde, Naofumi Takahashi, Yosuke Maeda, Akira Ono.

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
