## [Decision Letter · Decision Letter 0]

1 Jan 2026

PPATHOGENS-D-25-02833

M-Sec promotes the production of infectious HIV-1 virus through the exocyst complex

PLOS Pathogens

Dear Dr. Suzu,

Thank you for submitting your manuscript to PLOS Pathogens. After careful consideration, we feel that it has merit but does not fully meet PLOS Pathogens's publication criteria as it currently stands. Therefore, we invite you to submit a revised version of the manuscript that addresses the points raised during the review process.

The reviewers found your results of interest and potential importance, even though Reviewer 3 commented that the results are 'somewhat confirmatory' of your group's previous work.  However, all three reviewers have raised significant points of criticism, in particular on the number of cells imaged, the cell types used, the small molecule inhibitors used, and the size of the effects observed.  The reviewers therefore require a substantial amount of new experimental work.

We look forward to receiving your revised manuscript.

Kind regards,

Charles R M Bangham, ScD FRS

Academic Editor

PLOS Pathogens

Susan Ross

Section Editor

PLOS Pathogens

Sumita Bhaduri-McIntosh

Editor-in-Chief

PLOS Pathogens

orcid.org/0000-0003-2946-9497

Michael Malim

Editor-in-Chief

PLOS Pathogens

orcid.org/0000-0002-7699-2064

**Additional Editor Comments:**

The reviewers found your work of interest and potential importance, even though Reviewer 3 commented that the work is 'somewhat confirmatory' of your group's previous results. However, all three reviewers raise significant points of criticism, in particular regarding the number of cells imaged, the cell types used, and the size of the effects observed. All reviewers require substantial extra experimental work to be carried out; a new submission will therefore be required.

**Journal Requirements:**

At this stage, the following Authors/Authors require contributions: Reem M. Mahmoud, Masateru Hiyoshi, Randa A. Abdelnaser, Kazuaki Monde, Nami Monde, Takaaki Koma, Hidenobu Mizuno, Sara A. Habash, Naofumi Takahashi, Yosuke Maeda, Akira Ono, and Shinya Suzu. Please ensure that the full contributions of each author are acknowledged in the "Add/Edit/Remove Authors" section of our submission form.

2) Please amend your detailed Financial Disclosure statement. This is published with the article. It must therefore be completed in full sentences and contain the exact wording you wish to be published.

2) If any authors received a salary from any of your funders, please state which authors and which funders..

**Reviewers' Comments:**

Reviewer's Responses to Questions

**Part I - Summary**

Reviewer #1: Reem M. Mahmoud et al study how the cellular protein M-Sec regulates late steps of the HIV 1 life cycle and contributes to the production of infectious virions. Using knockdown in U87 cells, primary macrophages treated with the small-molecule M Sec inhibitor NPD3064, and M Sec overexpression in 293 cells, the authors show that M Sec promotes the formation and accumulation of HIV 1 Gag puncta, enhances Gag/Env co-localization, and increases Env incorporation into released particles without changing overall Env expression. There are modest changes in particle production but clear differences in infectivity, with M Sec–deficient cells producing less infectious virus and M Sec–overexpressing cells producing more infectious virus at equal RT-normalized input. Mechanistically, M Sec–dependent Gag puncta accumulation requires PIP2, the small GTPase Ral, and the exocyst complex (notably EXOC2 and EXOC3), suggesting that M Sec uses the same PIP2–Ral–exocyst axis previously implicated in tunneling nanotube formation to reorganize membranes and vesicular compartments that favor efficient Env incorporation into HIV 1 particles. The study is original and well executed. The authors propose that M Sec supports HIV 1 transmission through coordinated effects on cell structure and virion production.

Reviewer #2: HIV-1 infection of target cells can occur via cell-associated transmission modes or by cell-free particles. Previous work by the Suzu lab established that the M-Sec protein is involved in cell-associated transmission of HTLV and HIV-1. In the present study by Mahmoud and colleagues, the group analyzes the impact of M-Sec and production and infectivity of cell free HIV-1 particles. To this end, endogenous M-Sec expression is either reduced or M-Sec is ectopically overexpressed in virus cells that produce HIV-1 particles following transfection with proviral DNA or infection with HIV-1. The authors describe that reduced M-Sec expression reduces the accumulation of HIV-1 in intracellular puncta, while overexpression promotes the intracellular accumulation of Gag. Similar approaches also document a subtle effect of M-Sec on the recruitment of the viral glycoprotein Env to these intracellular accumulations and incorporation of Env into viral particles. Consistently, the presence of M-Sec has a moderate effect on the infectivity of these particles. Pharmacological inhibition of the Ral GTPase or EXOC7 abrogates the effects of M-Sec depletion or overexpression. The study addresses a relevant topic and presents interesting new data

Reviewer #3: The authors very recently reported in PLOS Pathogens that the host protein “M-Sec promotes the accumulation of intracellular HTLV-1 Gag puncta and the incorporation of Env into viral particles”. In the present study, they essentially repeated this study with HIV-1, and arrived largely at the same conclusions.

Specifically, they show that the depletion of M-Sec reduces, and exogenous M-Sec increases, intracellular HIV-1 Gag puncta, and that HIV-1 Env appears to co-localize with these Gag puncta. They also show that the M-Sec-mediated accumulation of Gag puncta depends on PIP2 (as it did in their previous paper on HTLV-I) and on Ral and the exocyst complex (whose SEC6 component exhibits sequence similarity with M-Sec). Additionally, they show that M-Sec promotes Env incorporation into viral particles and their infectivity. However, these effects were small (generally less than 2-fold), and whether these occur in relevant HIV-1 target cells was not examined.

Altogether, the present study appears to be somewhat confirmatory in nature, in the sense that it extends previous findings to yet another retrovirus. It is also difficult to reconcile with an earlier study by the same authors, in which M-Sec only affected HIV-1 particle production in the presence of Nef.

**Part II – Major Issues: Key Experiments Required for Acceptance**

Reviewer #1: 1. Experiments with primary macrophages are limited to a pharmacological inhibitor (NPD3064) effect on Gag puncta (S1). Given M-Sec is high in macrophages and low in CD4+ T cells, showing M-Sec knockdown (siRNA/ASO) in primary macrophages with effects on Env incorporation and infectivity would substantially strengthen the biological relevance. This should be at least further discussed.

2. Rescue of the U87 knockdown phenotype with shRNA-resistant M-Sec would address specificity.

3. Infectivity is normalized to RT activity; RT can vary with maturation. Is it possible to include normalization to p24 (ELISA)?

4. Assessing Gag processing/maturation in virions will allow to exclude M-Sec effects on protease-mediated maturation that could confound infectivity.

5. Can the authors provide quantification (densitometry, replicates) for EXOC2/EXOC3 knockdown efficiency and show that basal Gag puncta/Env colocalization in parental cells are affected similarly by exocyst knockdown.

6. It would be worth testing Gag MA basic patch mutants (defective in PIP2 binding) to determine whether M-Sec’s effect requires Gag–PIP2 interactions. This could be at least discussed

7. Image quantifications often use 10–15 cells per group, and only one or two cells are shown in the figures. What are the number of independent biological replicates? It may be worth showing lower magnification images with more cells.

Reviewer #2: I noted the following issues that should be addressed:

1) Most of the results in this study are deduced from microscopy analyses in which a few individual cells were analyzed (typically around 15). Although I note that this is how previous papers from the group have been published, this does not seem to be sufficient to draw meaningful conclusions. It is also not clear if these 15 cells were acquired in the same experiments or collected in different experiments. Considering that this is a straightforward analysis and cells to be analyzed are not a rare resource, analyzing at least three independent experiments with at least 20 cells per condition would seem a minimum. Also, the authors refer to previous papers for quality controls for the functional impact of M-Sec depletion on particle production. Since depletion is partial and stable cell lines may phenotypically adapt to the loss M-Sec expression, including this control as standard control in each experiment would be helpful.

2) The differences observed upon M-Sec depletion or overexpression in Gag relocalization, Env recruitment and virion incorporation, and particle infectivity are subtle (factor 2 or less). This does not mean that this is not an important phenomenon but it would need to be acknowledged more clearly as a very subtle phenotype. Maybe assessing the role of M-Sec over several rounds of replication would help to illustrate the relevance of the effect? Does the magnitude of infectivity alteration differ between Env variants with different tropism, tendency for shedding or other parameters?

3) Fig. 8: data on the infectivity of particles produced from cells treated with these inhibitors should be presented

Reviewer #3: 1. The authors show that M-Sec induces the accumulation of HIV-1 Gag puncta independently of other viral components (such as Nef). How can this finding be reconciled with their previously reported observation that in the absence of Nef M-Sec had no effect on HIV-1 virus production in a spreading infection assay (Fig. 5 of ref. 7).

2. The M-Sec-induced “Gag puncta” often are quite large, and appear to be mostly in a peri-nuclear location and rather than at the PM. What evidence do the authors have that these “Gag puncta” are not simply non-specific Gag aggregates, rather than assembling HIV-1 virions, as the authors seem to assume?

3. The conclusions largely rely on the use of shRNAs to stably knock down host proteins such as M-Sec in a cell line. However, in the experience of this reviewer, the stable expression of shRNAs frequently and significantly impairs HIV-1 replication in a manner that has nothing to do with the depletion of the intended target. It is therefore imperative that the authors perform rescue experiments.

4. U87 and 293 cells are not relevant HIV-1 target cells, and experiments with MDM are limited to the use of a poorly characterized M-Sec inhibitor. Depletion and rescue experiments should be performed with primary HIV-1 target cells to confirm the relevance of the observations.

**Part III – Minor Issues: Editorial and Data Presentation Modifications**

Reviewer #1: (No Response)

Reviewer #2: (No Response)

Reviewer #3: HIV-1 is primarily a T cell-tropic virus, but the authors state in the Discussion that M-Sec expression is low in primary CD4+ T cells and does not contribute to HIV-1 transmission. This is in agreement with the fact that certain human CD4+ T cell lines express no M-Sec at all, but are nevertheless highly permissive for HIV-1. In my opinion, it should be clarified in the title and abstract that the findings only apply to a subset of physiologically relevant target cells, if at all.

PLOS authors have the option to publish the peer review history of their article (what does this mean?). If published, this will include your full peer review and any attached files.

Reviewer #1: No

Reviewer #2: No

Reviewer #3: No

**Figure resubmission:**
---

## [Decision Letter · Decision Letter 1]

23 Mar 2026

PPATHOGENS-D-25-02833R1

M-Sec promotes the production of infectious HIV-1 virus through the exocyst complex in macrophages

PLOS Pathogens

Dear Dr. Suzu,

Thank you for submitting your manuscript to PLOS Pathogens. After careful consideration, we feel that it has merit but does not fully meet PLOS Pathogens's publication criteria as it currently stands. Therefore, we invite you to submit a revised version of the manuscript that addresses the points raised during the review process.

We look forward to receiving your revised manuscript.

Kind regards,

Charles R M Bangham, ScD FRS

Academic Editor

PLOS Pathogens

Susan Ross

Section Editor

PLOS Pathogens

Sumita Bhaduri-McIntosh

Editor-in-Chief

PLOS Pathogens

orcid.org/0000-0003-2946-9497

Michael Malim

Editor-in-Chief

PLOS Pathogens

orcid.org/0000-0002-7699-2064

**Additional Editor Comments :**

The reviewers appreciated your response to the points raised. Regarding the remaining points, in particular the first comment by reviewer 2, I would ask you to add the data from the other two experiments as requested: this could perhaps be added in supplementary data. However, I conclude that no new experiments are required at this stage.

**Reviewers' Comments:**

Reviewer's Responses to Questions

**Part I - Summary**

Reviewer #1: the authors have addressed my concerns

Reviewer #2: Mahmoud et al submit a revised version of their manuscript on the effect of M-Sec on the production and infectivity of HIV-1 particles. They made modifications in response to my previous comments and I will only address those comments that were not yet fully addressed.

Reviewer #3: In this revised paper, the authors have made a serious effort to address my concerns. For instance, they have performed a knockdown and rescue experiment in U87 cells to confirm that M-Sec induces the formation of Gag puncta in these cells. However, they did not examine whether the re-expression of M-Sec rescued Env incorporation and viral infectivity. I feel that these issues should also be examined in knockdown and rescue experiments, because the promotion of infectious particle production by M-Sec (not just the induction of Gag puncta) is what the paper really is about.

**Part II – Major Issues: Key Experiments Required for Acceptance**

Reviewer #1: (No Response)

Reviewer #2: 1.1: I appreciate that the number of cells analyzed was increased from 15 and 20. The authors also state that they show one representative of three experiments conducted essentially throughout the manuscript. The rationale for this is not clear to me – showing the results from all three replicate experiments for all the panels listed would seem good practice and will significantly add to the robustness of the study.

1.2: Thanks for adding this helpful experiment. This shows that the effect of M-Sec ko on Gag clustering can be rescued but the questions was regarding particle production. Please add the particle production/infectivity data from this experiment.

2.1.: The authors are now much clearer regarding the limited magnitude of their effects but opt against conducting a replication experiments over multiple rounds. I strongly encourage the authors to add data from such an experiment. It is simple to do and will either shown that your effects potentiates or vanishes over multiple rounds, which is helpful information either way.

Reviewer #3: (No Response)

**Part III – Minor Issues: Editorial and Data Presentation Modifications**

Reviewer #1: (No Response)

Reviewer #2: (No Response)

Reviewer #3: (No Response)

PLOS authors have the option to publish the peer review history of their article (what does this mean?). If published, this will include your full peer review and any attached files.

Reviewer #1: No

Reviewer #2: No

Reviewer #3: No

**Figure resubmission:**
---

## [Editor Report · Decision Letter 2]

22 May 2026

Dear Dr Suzu,

We are pleased to inform you that your manuscript 'M-Sec promotes the production of infectious HIV-1 virus through the exocyst complex in macrophages' has been provisionally accepted for publication in PLOS Pathogens.

Best regards,

Charles R M Bangham, ScD FRS

Academic Editor

PLOS Pathogens

Susan Ross

Section Editor

PLOS Pathogens

Sumita Bhaduri-McIntosh

Editor-in-Chief

PLOS Pathogens

orcid.org/0000-0003-2946-9497

Michael Malim

Editor-in-Chief

PLOS Pathogens

orcid.org/0000-0002-7699-2064

The reviewers consider that the manuscript is significantly stronger as a result of the recent changes; you may consider their remaining comments in future work.
---

## [Editor Report · Acceptance letter]

Dear Dr Suzu,

We are delighted to inform you that your manuscript, "M-Sec promotes the production of infectious HIV-1 virus through the exocyst complex in macrophages," has been formally accepted for publication in PLOS Pathogens.

Best regards,

Sumita Bhaduri-McIntosh

Editor-in-Chief

PLOS Pathogens

orcid.org/0000-0003-2946-9497

Michael Malim

Editor-in-Chief

PLOS Pathogens

orcid.org/0000-0002-7699-2064